# Joint disease-specificity at the regulatory base-pair level

Pushpanathan Muthuirulan[1,9], Dewei Zhao[2,9], Mariel Young[1], Daniel Richard[1], Zun Liu[1], Alireza Emami[3], Gabriela Portilla[3], Shayan Hosseinzadeh [3], Jiaxue Cao[1,8], David Maridas[4], Mary Sedlak[1], Danilo Menghini[3], Liangliang Cheng[2], Lu Li[2], Xinjia Ding[5], Yan Ding [6], Vicki Rosen [4], Ata M. Kiapour [3✉] & Terence D. Capellini [1,7✉]

Given the pleiotropic nature of coding sequences and that many loci exhibit multiple disease associations, it is within non-coding sequence that disease-specificity likely exists. Here, we focus on joint disorders, finding among replicated loci, that *GDF5* exhibits over twenty distinct associations, and we identify causal variants for two of its strongest associations, hip dysplasia and knee osteoarthritis. By mapping regulatory regions in joint chondrocytes, we pinpoint two variants (rs4911178; rs6060369), on the same risk haplotype, which reside in anatomical site-specific enhancers. We show that both variants have clinical relevance, impacting disease by altering morphology. By modeling each variant in humanized mice, we observe joint-specific response, correlating with *GDF5* expression. Thus, we uncouple separate regulatory variants on a common risk haplotype that cause joint-specific disease. By broadening our perspective, we finally find that patterns of modularity at *GDF5* are also found at over three-quarters of loci with multiple GWAS disease associations.

[1] Department of Human Evolutionary Biology, Harvard University, Cambridge, MA, USA. [2] Department of Orthopedics, Affiliated Zhongshan Hospital of Dalian University, Dalian, China. [3] Department of Orthopaedic Surgery, Boston Children's Hospital, Harvard Medical School, Boston, MA, USA. [4] Department of Developmental Biology, Harvard School of Dental Medicine, Boston, MA, USA. [5] Department of Surgery, the Second Affiliated Hospital of Dalian Medical University, Dalian, China. [6] Department of Pediatrics, Boston Children's Hospital, Harvard Medical School, Boston, MA, USA. [7] Broad Institute of MIT and Harvard, Cambridge, MA, USA. [8] Present address: Farm Animal Genetic Resources Exploration and Innovation Key Laboratory of Sichuan Province, Sichuan Agricultural University, Chengdu, China. [9] These authors contributed equally: Pushpanathan Muthuirulan, Dewei Zhao. ✉email: ata.kiapour@childrens.harvard.edu; tcapellini@fas.harvard.edu

The extent to which coding versus non-coding sequence variants underlie complex disease specificity and pathophysiology has been subject to much investigation[1,2]. Historically, because of their visibility, predicted effect, and relative ease in functional validation[3] the search for causal coding mutations has taken precedence over non-coding variants. However, non-coding variants are thought to have more regulatory specificity and therefore likely more modularized phenotypic impacts[4] making them in theory better candidates for mediating disease-specificity. In support of this concept, genome-wide association studies (GWAS) have consistently revealed that most common disease association signals are enriched in functional non-coding sequences[5–7], making the disease-causing variants more likely regulatory in nature.

Recently, with the development of phenome-wide association studies (PheWAS), which study the associations between genetic variants and many phenotypes[8], it has become apparent that many GWAS loci now reproducibly associate with multiple distinct diseases[9,10], causing these loci to have broader impacts on human health and disease. As these multiple disease-associated GWAS loci can contain many types of variants[8,9,11], disease-specificity at each locus may result from the actions of independent, but linked, causal coding mutations or regulatory variants. On the other hand, disease risk could also reflect the effects of a single pleiotropic variant that resides in coding or non-coding sequences. Addressing this issue using existing data on true allelic causality is not possible because compared to the number of known causal coding mutations, the number of bonafide causal non-coding variants (at hundreds of thousands of GWAS loci) is likely <20, despite advances in non-coding variant discovery[12] and modeling[13]. While we expect a portion of disease-specificity at multiple disease-associated GWAS loci to be under the effects

of spatiotemporally specific non-coding regulatory variants[14], clearly illustrated examples of such are lacking.

To begin to address this issue, we focused on the most common musculoskeletal diseases in the world, those affecting joints[15,16], in part because while they are often multifactorial in etiology, joint diseases can have sizable genetic underpinnings. For example, developmental dysplasia of the hip (DDH), a structural hip disorder present in 1 of 1000 newborns, has an estimated heritability of ~55%[17], while knee osteoarthritis (OA), a degenerative disease affecting >30% of people over 65 years of age, has a heritability of ~40%[18]. Why these and other joint diseases are both highly prevalent and heritable remains unclear but likely reflects in part the effects of genetic loci on the development of joint anatomy and morphology, biomechanical function, and homeostasis[19,20]. To identify loci that exhibit repeated connections to a myriad of joint diseases, thereby significantly impacting joint disease heritability broadly, we amassed datasets from GWAS on musculoskeletal synovial joint diseases and related traits (Supplementary Fig. 1), and identified 106 such loci. Of these GWAS loci, variants in *Growth Differentiation Factor Five* (*GDF5*), a gene with well-described roles in joint formation[21–29], associate via GWAS with at least thirteen separate skeletal and joint-related diseases and traits (e.g., DDH, hip and knee OA, joint hypermobility, knee pain, etc.), with another eight distinct diseases showing detectable *GDF5* signals via candidate gene association studies (Fig. 1, Supplementary Table 1; Supplementary Data 1 [17,30–45]). These findings make *GDF5* one of the most replicated inherited genetic risk factors for a myriad of distinct joint and skeletal diseases.

Given *GDF5* broad association with joint disease and that its coding mutations are well known to cause syndromic human phenotypes[21–29], it is expected that variants underlying its many GWAS associations are coding in nature and therefore act pleiotropically to cause multi-joint pathology. However, as alluded to there are no protein-coding mutations that can explain its population-level GWAS associations[46,47], and recent association studies and numerous clinical observations suggest instead a highly modularized joint-specific response at *GDF5*. Therefore, common variants at this locus, which span a >100 kb risk haplotype containing much non-coding sequence[17,30–45,48], likely have very specific impacts on individual joints, albeit in general, less is known regarding the causal variants at *GDF5* underlying its multiple GWAS associations[17,20,30–45].

Here, we address the issue of regulatory disease-specificity versus pleiotropy at the locus and genome-wide levels. First, at the *GDF5* locus we primarily focus on its two most common replicated disease associations, DDH[17,49] and knee OA[50]. Using a combination of functional genomics on human skeletal tissues, targeted genetics in the mouse model, assessment of patient genotypic data, relevant clinical measurements on imaging data from human patients and humanized mice, and previous insights, we demonstrate for each disease that genetic variants on the same *GDF5* risk haplotype can be functionally uncoupled and impact bone shape, and disease etiology, in a joint-specific manner. Second, we examine genome-wide whether signals of modularity at the chromatin accessibility level suggest disease specificity (as at *GDF5*) or pleiotropy for GWAS loci associated with multiple musculoskeletal traits and diseases. We find evidence in support of the effects of modularity on disease-specificity at the joint-specific level.

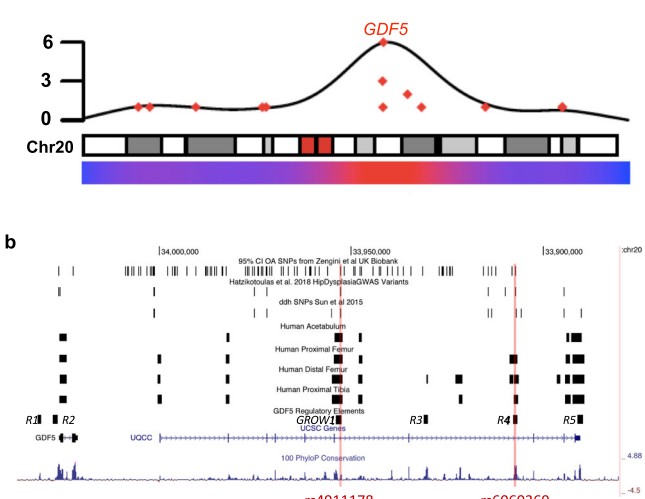

**Fig. 1 Musculoskeletal GWAS signals at the *GDF5* locus and intersection of regulatory sequences with the DDH and knee OA GWAS risk variants. a** Karyotype showing GWAS association of *GDF5* with joint diseases. The strongest signal on chr20 is in the *GDF5-UQCC1* locus (seven significant SNPs; see Supplemental section for more details). **b** Modified UCSC Genome Browser view (hg19) showing the intersection of DDH[17,49] and knee OA[50] risk variants (top) with Human ATAC-seq regions (from acetabulum, proximal femur, distal femur, proximal tibia tissues) (middle), *GDF5* regulatory elements (*R1–R5*, *GROW1*)(bottom), followed by UCSC gene locations and peaks of phyloP100ways conservation. Two variants, rs4911178 in *GROW1*, and rs6060369 in *R4*, overlap with functional regulatory sequences in embryonic human tissues and mouse (see also Supplementary Fig. 2).

## Results

**Disease risk variants in chondrocyte regulatory regions.** GWAS across a number of common joint diseases and traits consistently detect a *GDF5* risk haplotype spanning approximately >100 kb of

mainly non-coding sequence. Moreover, to date, no protein-coding mutations have been found that can explain these GWAS associations (Supplementary Fig. 1 and Supplementary Data 1 in the Supplementary Information)[46,47]. These findings point to non-coding regulatory variants as likely causal for each disease risk at the locus, yet they remain mostly unknown, and importantly it is unclear whether GDF5-mediated joint diseases share a common pleiotropically acting causal regulatory variant, or not. We therefore sought to identify causal variants at GDF5 by focusing on its most replicable GWAS disease associations, DDH and knee OA.

Both DDH and knee OA are diseases that arise in part from altered chondrocyte biology[51–53]. Moreover, DDH is well known to result from altered prenatal hip development (reviewed in Wilkinson et al.,[19]) while knee OA heritability results in part from genetic variants altering developmental chondrocyte knee enhancer function[20]. Given these insights, we performed ATAC-seq, an assay that detects accessible chromatin regions genome-wide[54,55], on chondrocytes acquired from developing gestational day E67 human embryonic hips (acetabula, proximal femora) and knees (distal femora, proximal tibiae) to identify joint-specific non-coding regulatory sequences (Supplementary Data 2; see Supplementary Information). Besides logistical issues in sample acquisition, this time-point was chosen because it is a stage when both joints and their respective morphologies are actively forming, and when their chondrocyte rudiments are relatively homogenous in cellular composition[19,20,56] (see Supplementary Information). ATAC-seq regions from each embryonic structure were also intersected with ATAC-seq regions acquired from chondrocytes from the same structures from approximate stage-matched E15.5 mouse embryos (Supplementary Data 2) in order to assess the relative functional orthology/conservation of any regulatory sequence but also to inform on whether the mouse could serve as an effective in vivo system to model variant effects. In addition, as altered chondrocyte biology during aging leads to knee OA, we examined a previously published ATAC-seq dataset from OA patient knees[57]. All human embryonic/adult and mouse embryonic ATAC-seq datasets were next intersected with DDH[17,49] and knee OA[50] GWAS risk variants (Supplementary Data 3; see Supplementary Information). We discovered two (rs4911178 and rs6060369) variants that consistently overlapped functional developmental sequences in both humans and mouse, and overlapped with signals in the aging OA knee (Fig. 1 and Supplementary Fig. 2 in the Supplementary Information). A discussion on other ATAC-seq/variant intersections is provided in Supplemental Information. Below, we functionally characterize these two variants in human patients and mouse models to show their phenotypic impacts and address whether they have pleiotropic effects driving both diseases or separate independent effects on hip versus knee morphology and disease pathophysiology.

**rs4911178 and DDH in humans and mice**. The variant rs4911178 (G → A) resides in an ATAC-seq accessible chromatin region that was previously reported as GROW1, a regulatory enhancer active in human and mouse long bone and proximal femoral neck growth plate chondrocytes but inactive in the knee joint proper (i.e., distal femur and proximal tibia epiphyseal or articular chondrocytes)[48]. Given the enhancer's anatomical- and cellular-specificity as well as functional conservation, we first assessed the clinical relevance of the "A" risk allele to DDH by examining its prevalence in 113 DDH patients (Supplementary Data 4). In patients, the "A" allele frequency was 84.5%, a highly significant enrichment (e.g., 8.3% increase) compared to population controls at 76.2% ($p < 0.005$). Moreover, the homozygous

risk "A/A" genotype frequency was 73.5%, a highly significant enrichment (e.g., 16.2% increase) compared to controls at 57.3% ($p < 0.005$), and of the 113 patients, 83 were homozygous risk "A/A" while only five individuals harbored the non-risk "G/G" genotype (Supplementary Data 4). We next considered the morphological impacts of the "A" allele in this patient dataset, finding that "A/A" individuals had a more vertically aligned acetabulum (and other modifications) compared to "A/G" individuals (Fig. 2a), indicating that the allele is associated with increased severity within DDH patients. These acetabular modifications are widely recognized hallmarks for DDH, and are features that differ compared to normal controls from China, Japan, and Korea (Supplementary Fig. 3).

These findings in DDH patients prompted us to model the GROW1 enhancer and rs4911178. "A" risk variant in mouse and human chondrocytes to gauge their respective impacts on hip versus knee biology. First, we generated a novel GROW1-LacZ reporter mouse line driven by the human version of the enhancer and found that it drives expression in growth plate chondrocytes of the acetabulum and proximal femoral neck at E15.5 (Fig. 2b), corroborating the approximate stage- and tissue-matched E67 (and independently, E59[20]) human and E15.5 mouse ATAC-seq signals (Fig. 1). Expression was also present in the distal femur, albeit restricted to growth plate chondrocytes above, and not within, the knee joint proper (Fig. 2b). Second, using an existing GROW1 enhancer null mouse line[48], we analyzed clinically relevant morphological features of the hip and knee, and only found significant changes in acetabulum and proximal femur growth plates (Fig. 2c) (see also Supplementary Fig. 3 and 4, and Supplementary Tables 2–8 in the Supplementary Information). The observed morphological differences include a vertically oriented acetabulum, and other changes, as seen in human DDH patients (Fig. 2a; Supplementary Fig. 3). Third, to gauge the impacts of enhancer function in human chondrocytes, we used CRISPR-Cas9 editing and deleted GROW1 and separately a 13 bp sequence containing rs4911178 and found each excision significantly reduced GDF5 expression levels (Fig. 3a and b), albeit no changes to nearby gene expression were observed.

Given previous studies on the "A" risk variant's relationship to height variation[48] and that DDH and knee OA are complex three-dimensional phenotypes that develop over months to years, we engineered for this study humanized single allelic replacement "A" mice at the orthologous rs4911178 position (i.e., GROW1^{rs4911178-A/rs4911178-+} mice) (see Supplementary Information). Allele-specific expression (ASE) assays on these mice revealed that the presence of just one "A" risk allele significantly reduced Gdf5 expression in vivo in proximal femur chondrocytes (~16% reduction in Gdf5 expression when normalized by wild type expression; $p = 0.0098$; $n = 7$ per genotype), a stronger expression reduction observed than in growth plate chondrocytes from the distal femur (Fig. 3c), therefore suggestive of a specific and major regulatory impact in vivo during hip development.

To understand the molecular mechanism of action for this expression change, chromatin immunoprecipitation (ChIP) was next performed on human chondrocytes (T/C-28a2 cells) in vitro and on engineered single base-pair replacement mice in vivo for the major hind limb transcription factor PITX1[58–60]. We found significant PITX1 binding at GROW1 in both human and mouse contexts (Fig. 3d and e; see Supplementary Information). Moreover, the "A" variant's impact on expression is directly mediated by changes in PITX1 binding—i.e., the "A" variant binds PITX1 more weakly than the conserved "G" non-risk variant in vivo with an even weaker binding in the hip compared to that in the distal femoral growth plates (Fig. 3f). We next performed PITX1 over-expression experiments in mouse embryonic (E15.5) chondrocytes genotypic for

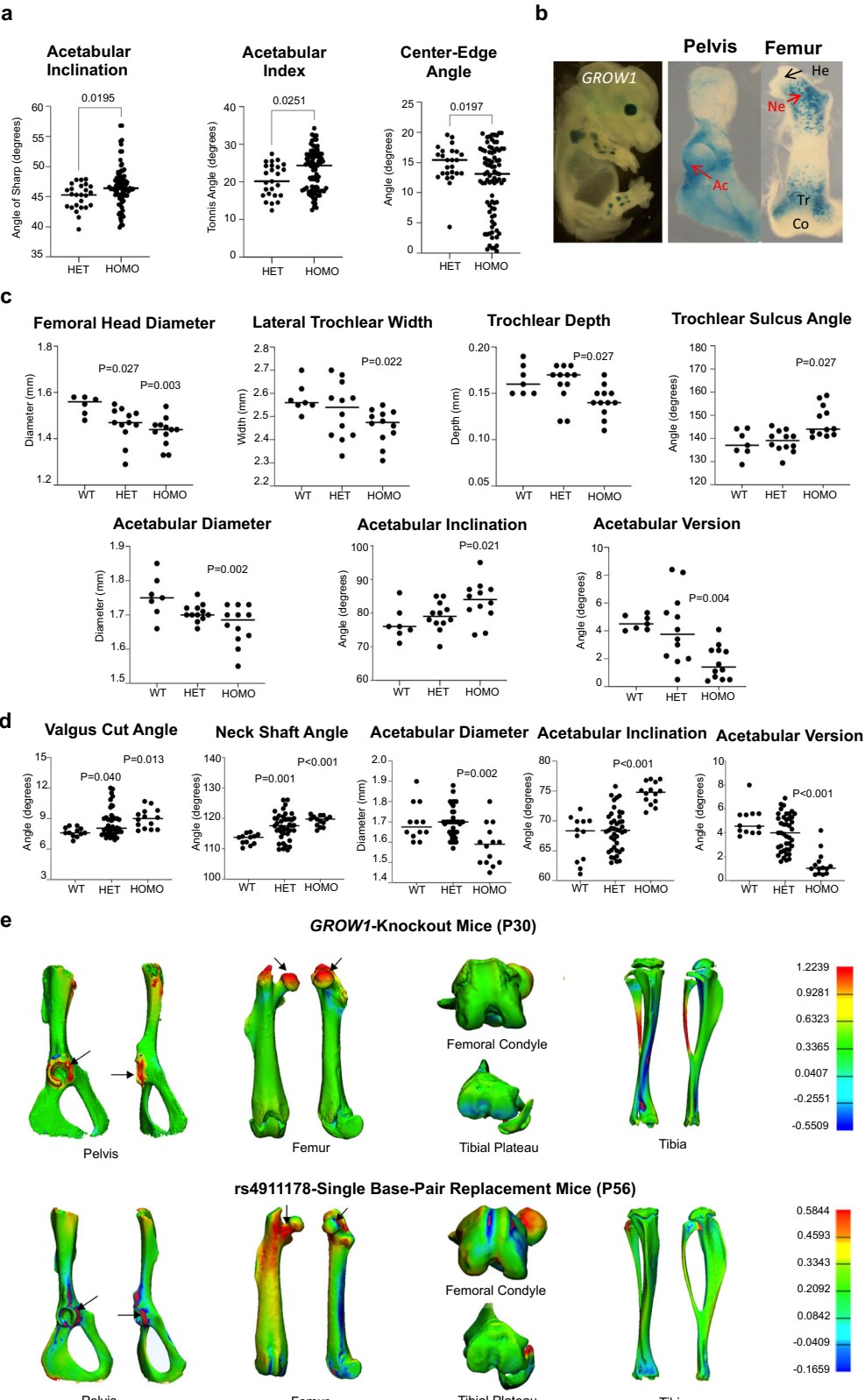

*GROW1rs4911178-A/rs4911178-G*, harvested from the proximal femur, distal femur, and proximal tibia growth plates, to identify how PITX1 influences *Gdf5* expression in different structures of the hind limb. The use of the PITX1-overexpression construct greatly increased the intracellular levels of *PITX1* mRNA in chondrocytes derived from the proximal femur (9.30-fold, $n = 3$,

$p = 0.0034$), distal femur (9.92-fold, $n = 3$, $p = 0.0114$), and proximal tibia (6.03-fold, $n = 3$, $p = 0.013$) (Fig. 3g), which resulted in stimulated response for *Gdf5* mRNA of 7-fold ($n = 3$, $p = 0.004$), 5.42-fold ($n = 3$, $p = 0.023$) and only 0.75-fold ($n = 3$, $p = 0.683$) for the proximal femur, distal femur and proximal tibia, respectively (Fig. 3g). Using MiSeq on pulled-down

**Fig. 2 Morphological characterization of the *GDF5 GROW1* enhancer and rs4911178 variant in patients and the mouse model. a** Higher acetabular inclination angle and acetabular index along with lower center-edge angle in homozygous "A/A" risk DDH patients compared to heterozygous "A/G" controls (HET $n = 25$, HOMO $n = 83$). Independent $t$-tests with two-sided $p$-values were used. Bars indicate medians. **b** *GROW1*-driven *lacZ* expression in growth zone chondrocytes of the acetabulum and proximal femoral neck and trochanteric regions at E15.5. Acetabulum (Ac), Head (He), Neck (Ne), Trochlea (Tr) and Condyles (Co) (red text denotes locations of expression, black text indicates no expression). **c** μCT measurements of significantly affected anatomical features in *GROW1* null mice at postnatal day, P30 (WT $n = 7$, HET $n = 12$, HOMO $n = 12$). ANOVA with Dunnet post-hoc was used for pairwise comparisons to wild type. All $p$-values are two-sided and indicate significant differences compared to wild type. Bars indicate medians. **d** μCT measurements of significantly affected anatomical features in rs4911178 single base-pair replacement mice at postnatal day, P56 (WT $n = 12$, HET $n = 42$, HOMO $n = 14$). ANOVA with Dunnet post-hoc was used for pairwise comparisons to wild type. All $p$-values are two-sided and indicate significant differences compared to wild type. Bars indicate medians. **e** 3D comparative analysis indicating the anatomical locations of largest morphological differences between wild type *GROW1*$^{+/+}$ and homozygous *GROW1*$^{-/-}$ hind limbs (top) as well as between wild type *GROW1*$^{rs4911178-G/rs4911178-G}$ and homozygous risk *GROW1*$^{rs4911178-A/rs4911178-A}$ hind limbs (bottom). The areas with the largest variations are highlighted in red (wild type > homozygous) and dark blue (wild type < homozygous), with minimal variations displayed in green/yellow. Bars indicate medians. All $p$-values are two sided. See Supplementary Figs. 3 and 4, Supplementary Tables 2–8 and the Supplementary Information for related analyses.

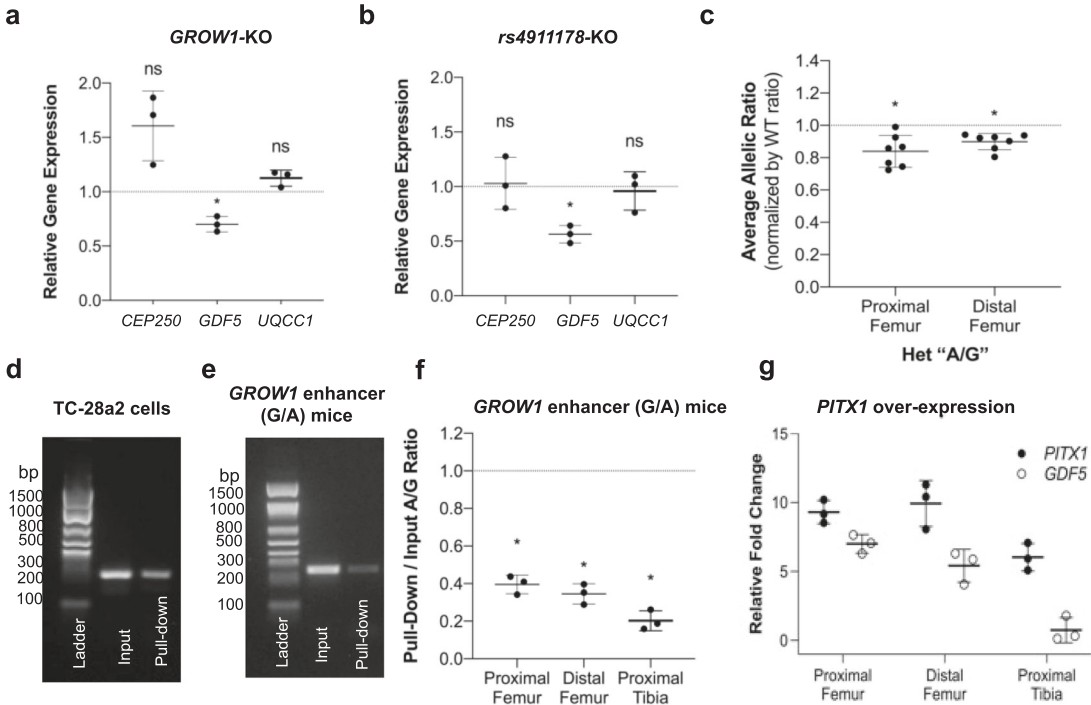

**Fig. 3 Molecular characterization of the *GDF5 GROW1* enhancer and rs4911178 variant in human chondrocytes and the mouse model. a** Expression of *CEP250* (non-significant (0.082)), *GDF5* (significant, $p < 0.018$), and *UQCC1* ($p = 0.099$) in human T/C-28a2 chondrocytes lacking the *GROW1* enhancer ($n = 3$ biologically independent samples). **b** Expression of *CEP250* ($p = 0.849$), *GDF5* ($p = 0.011$), and *UQCC1* ($p = 0.731$) in T/C-28a2 cells lacking a 13 bp region containing rs4911178 in *GROW1* ($n = 3$ biologically independent samples). **c** ASE assays on E15.5 proximal ($p = 0.019$) and distal femur ($p = 0.038$) chondrocytes from C57BL/6 J/129SvJ *GROW1*$^{rs4911178-A/rs4911178-+}$ replacement mice ($n = 7$ biologically independent samples). **d** PITX1 ChIP assay on a 234 bp *GROW1* sub-region containing rs4911178 in human T/C-28a2 chondrocytes showing input (left) and pull-down (right) in the gel image. This experiment was repeated three times independently with similar results. **e** PITX1 ChIP assay on a 255 bp *GROW1* sub-region containing rs4911178 in *GROW1*$^{rs4911178-A/rs4911178-G}$ mice showing input and pull-down from E15.5 proximal femur. This experiment was repeated three times independently with similar results. **f** A/G allelic ratio in PITX1 ChIP pull-down compared to input from *GROW1*$^{rs4911178-A/rs4911178-G}$ mice (39.6% decrease in 'A' over G, $p = 0.002$ in the proximal femur; 34.5% decrease in 'A' over 'G', $p = 0.002$ in distal femur; 20.2% decrease in 'A' over 'G', $p = 0.001$ in proximal tibia) ($n = 3$ biologically independent samples). **g** Fold-change for *PITX1* and *Gdf5* gene expression in PITX1 over-expressed chondrocytes cultured from E15.5 embryonic hind limb buds (proximal femoral, distal femoral, and proximal tibia). Summary data are presented as mean and standard deviations. All $p$-values are two-sided.

amplicons from this PITX1 over-expression experiment, we found there was an enhanced allelic imbalance whereby the overexpression of PITX1 also increased the differential binding of "G" versus "A" allele as anticipated (4.136-fold increased binding of "G" allele to PITX1) in the proximal femur. Overall, these data demonstrate that the "A" risk variant at rs4911178 directly decreases PITX1 binding, revealing a molecular mechanism of action of the variant change at the specific variant and most markedly in the proximal femur.

Next, to understand the impacts of the "A" variant on skeletal morphology, morphometric analyses were conducted on post-natal single base-pair replacement mice (see Supplementary Information). We observed that homozygous risk *GROW1*$^{rs4911178-A/rs4911178-A}$ mice have clinically relevant DDH phenotypes compared to controls; strikingly, alterations to the orientation of the acetabulum (e.g., a more vertically oriented acetabulum as we also observed in "A/A" DDH patients compared to "A/G" patients) and femoral neck (Fig. 2d, and

Supplementary Tables 2–8 in the Supplementary Information), with no detectable changes within the knee joint proper (Supplementary Figs. 3 and 4, and Supplementary Tables 2–8 in the Supplementary Information). The specificity of the hip phenotypes is most evident via morphological heatmaps of both engineered enhancer null and single base-pair homozygous risk "A/A" replacement mouse lines compared to wild type controls (Fig. 2e). Collectively, these data reveal that GROW1 and more importantly its "A" risk variant at rs4911178 affect the hip and cause clinically relevant hip phenotypes as observed directly in DDH patients, but do not impact knee morphology.

**rs6060369 in knee biology and OA in humans and mice.** The variant rs6060369 (C → T) resides in an ATAC-seq region that was previously reported as R4, a strong regulatory enhancer active in embryonic and adult human and mouse knee epiphyseal and articular chondrocyte joint territories, but only weakly active in the hip and absent from growth plate chondrocytes[20]. These findings match a similar time-course in human knee chondrocytes, as revealed here by E67 chromatin accessibility ATAC-seq signals (Fig. 1; Supplementary Fig. 2 and Supplementary Data 2 in the Supplementary Information) and signals observed in human knee OA patient cartilage (Supplementary Data 3). Given its replicable association with knee OA[20], we next sought to examine the behavior of the "T" risk allele in the Osteoarthritis Initiative (OAI) (see https://oai.epi-ucsf.org) dataset consisting of 4,796 men and women, with different extent of radiographic knee OA as quantified by standard Kellgren and Lawrence (KL) grades, followed prospectively and longitudinally over 10 years (Clinical Trials.gov - NCT00080171). We asked whether patients with no radiographic knee OA (KL = 0/1) over the duration of the study possessed greater or lesser "T" variants than those with radiographic knee OA (KL = 2/3) at baseline, or those developing radiographic OA (progressing to KL ≥ 2) by the time of their last follow-up (see "Methods"). We found the "T" allele was lower in individuals who never progressed compared to those who have KL ≥ 2 OA status at baseline ($p = 1.9e{-}11$) or progressed to KL ≥ 2 ($p = 4e{-}4$), indicating that the "T" allele mediates knee OA onset and/or progression (Fig. 4a). In order to put these findings in the context of (any observable) morphological alterations mediated by the "T" allele, we also quantified several key morphological features of the knee joint from baseline MRI of the patients with KL = 0 (n = 383). We observed significant effects on knee size as well as condylar curvature, with "T/T" genotype associated with smaller femoral condyles and tibial plateau, and more curved medial femoral condyles (Fig. 4b).

We previously showed that the human risk "T" variant when modeled directly in mice (i.e., in R4$^{rs6060369-T/rs6060369-T}$ allelic replacement mice) impacts Gdf5 expression and knee shape early in life with a potential role later. However, whether the allele causes more drastic shape defects and OA later in life remained unstudied, and it was unknown if it impacts hip biology. Using micro-CT imaging on 1-year old homozygous risk R4$^{rs6060369-T/rs6060369-T}$ allelic replacement and wild type mice, we now have found that the "T" allele causes significant and marked changes to the width and curvature of the knee's femoral condyles as well as to the size and slopes of the tibial plateau, with similar patterns as those seen in OA patients with "T" risk alleles (see above), and in R4$^{-/-}$ mice (Fig. 4c, Supplementary Table 9). These altered anatomical features include those highly correlated to OA severity at 1-year[20].

To move past correlation and show direct causality of the "T" allele in knee OA, we next performed histological analysis and osteoarthritis (OARSI) scoring on knees of R4$^{rs6060369-T/rs6060369-T}$ allelic replacement and littermate wild type mice. Indeed, by 1 year of age we observed that R4$^{rs6060369-T/rs6060369-T}$ mice exhibit

significant knee OA (Fig. 4d–e). Similar to that observed in enhancer null (R4$^{-/-}$) mice at 1 year[20], R4$^{rs6060369-T/rs6060369-T}$ mice display varying OARSI scores (Fig. 4e), with some mice showing no signs of histological OA whilst others showing severe OA. In mildly affected individuals we observe minor cartilage tears, whereas in strongly affected individuals we observed major cartilage loss (Fig. 4d). These data support the hypothesis that knee shape alterations, via this GDF5 "T" variant, precede observable knee OA. Moreover, these data reveal that modeling single-nucleotide non-coding disease risk variants in the mouse can directly reveal functional causality for complex diseases.

As the "T" allele at the R4 enhancer also resides on the same DDH risk haplotype (harboring rs4911178) and therefore could be causal for hip disease, we assessed both R4 null and "T" risk allelic replacement mice at P30 and 1-year focusing on the proximal femur and acetabular morphology (see Supplementary Information). No hip modifications in homozygous or heterozygous risk genotypes for either model were found (Fig. 4f; Supplementary Figs. 5–8 and Supplementary Tables 10–15 in the Supplementary Information), also evident via morphological heatmaps of both R4$^{-/-}$ and R4$^{rs6060369-T/rs6060369-T}$ mice compared to wild type controls. Thus, the rs6060369 risk "T" variant, as well as loss of the R4 enhancer itself, have no observable impacts on hip morphology in mice, indicating a prime role for the "T" allele directly in knee shape and OA pathogenesis.

**Functional impacts of GDF5 variants in complex trait and disease.** GWAS associations and our functional studies above indicate that the highly localized control of GDF5 expression level is important for a myriad of human joint disease risks. To gauge this better, we performed a regression analysis to identify how loss-of-function and single base-pair modifications influence Gdf5 expression and phenotype in four engineered enhancer mouse lines described above. We observed a strong linear relationship between the level of Gdf5 expression and the extent of change in clinically relevant anatomical features, with a higher reduction in Gdf5 expression leading to larger morphological changes (Fig. 5a). Yet, this was evident for anatomical structures directly mediated by anatomically relevant enhancer modifications. On average, homozygosity at R4 or GROW1 variants resulted in 1.7 (±0.5) fold greater changes in anatomy compared to heterozygous mutations (Fig. 5b). Also, for every 1% reduction in Gdf5 expression, there was a 0.57% change in anatomy (Fig. 5a). Therefore, while expression levels of GDF5 appear critical for phenotypes, enhancer specificity, as revealed by our detailed functional genomic and genetic experiments in vivo in humans and mouse, ultimately underlies where in the body, and in which cell type, anatomy is impacted. Our findings are summarized in Fig. 5c.

**Broader patterns of modularity at musculoskeletal GWAS disease and trait loci.** We next took a broader approach to understand whether the modularity/disease specificity relationship we functionally dissected at GDF5 is unique to that locus or could be a more consistent pattern at musculoskeletal GWAS loci exhibiting multiple disease/trait associations. To carry out these studies, we first examined all loci genome-wide (i.e., those with or without musculoskeletal GWAS disease/trait signals) and then those loci exhibiting singular versus multiple musculoskeletal GWAS disease/trait-associated signals. For each locus we asked whether it exhibits a pattern of non-modularity or modularity in chromatin accessibility using our fetal chondrocyte ATAC-seq data from the pelvis, femur, and tibia (see Supplementary Information; Supplementary Data 1), noting the tissue-type and time-point limitations in this dataset. Briefly, we defined a locus

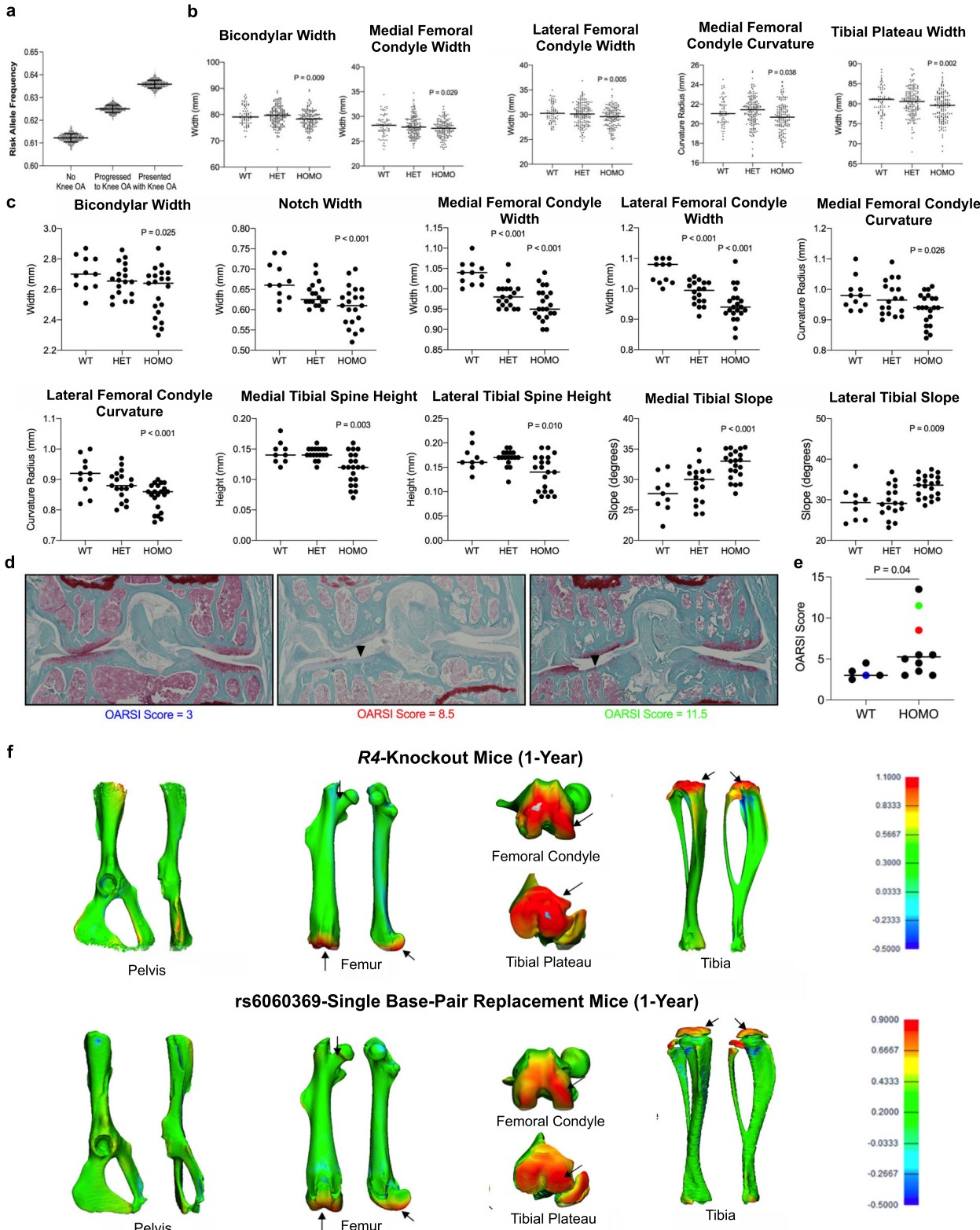

as being non-modular if, at each detected putative regulatory element position, the same tissues displayed an open chromatin site (e.g., at *ERCC1* we identified 13 regulatory regions, all of which show accessibility in all tissues; at *TMX* we identified nine regions which all have the same tissue accessibility representation) (Supplementary Data 1). We then defined loci that behave as modular as those where the representation across tissues varied at more than one regulatory element position across the locus

(e.g., at *GDF5* we identified seven distinct regions several displaying different tissue overlaps) (Supplementary Data 1). In this analysis, we found that genome-wide, 26,107 gene loci had ATAC-seq signals (from at least one tissue) within 100 kb of its TSS, understanding that without chromatin capture data this latter distance requirement likely inflates the number of loci. Of these, 21,486 (82.29%) gene loci behave as modular, versus 4,621 (17.71%) gene loci as non-modular. Of the non-modular gene

**Fig. 4 Morphological characterization of the *GDF5 R4* enhancer and rs6060369 variant in patients and the mouse model. a** Occurrence of 'T' risk allele frequency at rs6060369 in subsampled OAI patients stratified by OA progression (see text for p-values). Points represent individual sub-samples ($n = 200$) of individuals taken from patient groups presenting with no/moderate-OA (KL = 0/1) ($n = 1207$), progressing to significant OA ($n = 208$), and presenting with significant OA (KL $\geq$ 2) ($n = 1119$). A two-sided Student's *T*-test was used to compare risk-allele occurrence in subsamples of each group, with FDR correction for $n = 3$ tests. Central bar represents mean of distribution, lower/upper bars indicate first and third quartiles, respectively. **b** Significantly different anatomical features of the knee in OAI patients (WT $n = 76$, HET $n = 161$, HOMO $n = 146$). ANOVA with Dunnet post-hoc was used for pairwise comparisons to wild type. All p-values are two-sided and indicate significant differences compared to individual's wild type for the non-risk allele. Bars indicate medians. **c** µCT measurements of significantly different anatomical features in rs6060369 single base-pair replacement mice at 1-year (WT $n = 6$, HET $n = 15$, HOMO $n = 14$). ANOVA with Dunnet post-hoc was used for pairwise comparisons to wild type. All p-values are two-sided and indicate significant differences compared to wild type. Bars indicate medians. **d** Histological analysis of rs6060369 single base-pair replacement mice at 1 year, showing coronal sections at mid-knee level. The sections represent knees with mild, moderate, and severe knee OA. Sections are color coded to their corresponding overall score listed in the graph in (**e**). The scale bars are 250 µm. **e** OARSI scores on wild type and homozygous rs6060369 single base-pair "T" replacement knees at 1 year (WT $n = 5$, HOMO $n = 10$). Mann–Whitney test with two-sided p values were used for comparison. **f** 3D comparative analysis indicating the anatomical locations of largest morphological differences between wild type $R4^{+/+}$ and homozygous $R4^{-/-}$ hind limbs (top) as well as between wild type $R4^{rs6060369-A/rs6060369-A}$ and homozygous $R4^{rs6060369-T/rs6060369-T}$ hind limbs (bottom) at 1-year. All p-values are two-sided. See Supplementary Figs. 8–9, Supplementary Tables 10–15 and the Supplementary Information for related analyses.

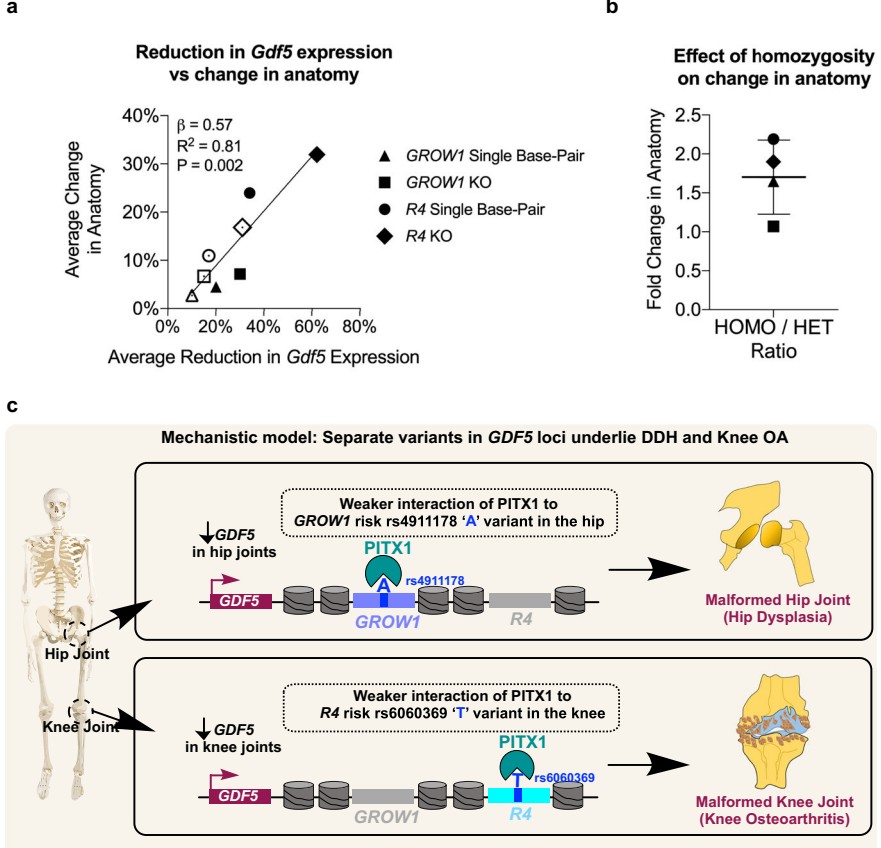

**Fig. 5 Association between *GDF5* expression and phenotype severity. a** Linear regression analysis showing a strong linear relationship between levels of *Gdf5* expression and extent of bony shape changes in four distinct mouse lines, two of which harbor separate enhancer deletions (i.e., of *R4* and *GROW1*) and two of which harbor separate, single-base-pair replacements in these enhancers (i.e., at rs6060369 and rs4911178). Dark circles = homozygous, open circles = heterozygous. See the Supplementary Information for related analyses. **b** Average fold change in anatomy between homozygous and heterozygous (enhancer null and single base-pair) mutations in *R4* and *GROW1* enhancers. Summary data are presented as mean and standard deviations. **c** Mechanistic model showing two separate non-coding regulatory variants on the same risk *GDF5* haplotype, i.e., rs4911178 "A" variant in the *GROW1* enhancer and rs6060369 "T" variant in the *R4* enhancer control different pathologies (i.e., DDH and knee OA, respectively) in different joints, mediated by differential binding of PITX1 on each respective enhancer variant in each specific joint.

loci, only 761 (2.91%) have complete pleiotropy (i.e., all elements across the locus exhibited accessibility in all five tissues examined), while the rest exhibit consistent overlap (i.e. shared patterns) with 1, 2, 3, or 4 tissues across all elements. Thus, gene loci in general display evidence of modularity (see "Discussion"; Supplementary Data 1).

We then examined those gene loci harboring any musculoskeletal GWAS signal, and found 825/955 (86.4%) behave as modular, which is a slight but significant enrichment compared to the genome-wide modular signal we observed above (hypergeometric p-value = 0.0002, 1.049 fold-change increase). Sub-setting on those loci exhibiting multiple musculoskeletal

GWAS disease/trait signals, we found that 77.4% (82/106) behaved as modular and only 22.6% (24/106) behaved as non-modular, with 6 that exhibited complete pleiotropy (Supplementary Data 1). Thus, more than three-quarters of multiple-disease/trait GWAS loci behave modularly, indicating that *GDF5* is not unique in this regard. By comparing these multiple disease/trait modularity findings to those examined at all musculoskeletal GWAS disease/trait loci, we found that this signal of modularity constituted a slight depletion (hypergeometric, $p = 0.002$, fold change $= 0.895$).

## Discussion

One key finding from this study is that separate non-coding regulatory variants on the same risk haplotype, i.e., rs4911178 "A" in the *GROW1* enhancer and rs6060369 "T" in the *R4* enhancer[20], can be functionally uncoupled to independently control different pathologies (i.e., DDH and knee OA) in different joints (Fig. 5b). In both cases, the master hind limb transcription factor, PITX1, operates hierarchically to mediate each risk variant effects on *GDF5* expression in vivo, but does so separately on the *GROW1* variant in hip growth plate chondrocytes, and the *R4* variant in knee epiphyseal and articular chondrocytes[20]. Thus, the common, high-frequency *GDF5* haplotype associated with many skeletal traits and diseases[48] (Supplementary Table 1) does not reflect the effects of one isolated pleiotropic causal coding mutation or even pleiotropic non-coding variant, but the weighted effects of different causal cis-regulatory variants on joint- and tissue-specific development and disease pathogenesis (Fig. 5b). We note that we cannot rule out the interacting effects of any additional cis-acting variants in influencing each specific disease at *GDF5*. Overall, these findings have important ramifications to how we think about disease etiology, therapeutics, and risk predictive strategies at this locus, as well as how we think about and investigate the architecture of complex traits and diseases (see below).

In DDH and knee OA patients, and as validated in our humanized mouse models, the observed *GDF5* enhancer mediated skeletal modifications led to normal-looking joints with small, yet clinically significant effects on key joint-specific morphological features, leading to altered biomechanics and stability. To this end, even global heterozygous *Gdf5* null mice lack proximal femur and distal femur dysmorphologies and have bones and joints that appear as modestly scaled-down versions of wild type forms[21,61–65]. Moreover, it is interesting that DDH and primary knee OA have not been reported in heterozygous (and for knee OA, even homozygous) *Gdf5* null mice[21,61–63,65]. Collectively, these findings indicate that it is the misregistration of local (i.e., joint-specific) anatomical relationships that are important drivers of different common disease outcomes at this locus, likely in the context of other genetic and non-genetic factors. This site-specific developmental effect, however, poses significant challenges when thinking about the development of GDF5-based therapies—i.e., they may only be effective at the early stages of development and growth, well before actual joint-specific pathology develops. Yet, the specificity of GDF5 effect also makes it a fruitful target for joint-specific intervention, as is (already) required in the treatment of most cases of joint disease, which are often localized and not systemic. Moreover, each variant's joint specificity should also aid in the early identification of such at-risk individuals which can be beneficial for developing personalized medicine approaches and preventative measures achieved by prophylactic interventions and lifestyle modifications. Lastly, this specificity may also be used to guide treatment, not only in understanding the target tissue under each variant's influence (e.g., femoral neck and acetabulum growth plate chondrocytes in DDH versus distal femur epiphyseal and articular chondrocytes in knee OA) but in particular in complex joint diseases (e.g., knee OA), where disease can be driven by several external factors, including past injury[66–68] and obesity[69–71] interacting with genetics.

As revealed through our examination of musculoskeletal GWAS disease/trait loci and our human skeletal element accessibility data, this built-in modularity at *GDF5* and its effects on different diseases may not be unique, but instead a common occurrence for loci recurrently impacting many different cartilage and bone diseases and traits. Indeed, we found that over three-quarters of loci that exhibit multiple musculoskeletal GWAS disease/trait signals may behave in a manner similar to *GDF5*, and thus these loci should be functionally interrogated with this in mind. However, it is also clear that modularity is likely the rule across loci exhibiting even singular GWAS signals (to date) as well as those yet showing evidence of association. We believe these relationships at musculoskeletal disease loci reflect the manner in which the musculoskeletal system develops and functions, but in the context of how evolution has shaped its anatomical site-specific regulation.

Prior to this work, it has been shown that genes involved in musculoskeletal development (e.g., *Bmp2*[72–74], *Bmp4*[75], *Bmp5*[76], *Gdf5*[61], *Gdf6*[77,78], *Acan*[79], among others) have complex, modularized regulatory systems, with different enhancers driving expression at distinct anatomical sites and/or tissue types. Several of these genes also appear in our analyses (see Supplementary Data 1). Evolutionarily, these and likely many other modularized regulatory gene loci evolved core functions in vertebrates for their roles in skeletal development and homeostasis. Namely, as skeletal system development requires the precise anatomical control of its different bones and joints, this built-in modularity likely arose as functional skeletal structures, such as joints, evolved to meet the different demands of movement on different substrates[80]. The evolutionary signatures of this control (i.e., functionally conserved modular elements), as well as the likelihood of some degree of regulatory element turnover at such loci[81], presents situations in which structures have the ability to evolve compartmentally, without impacting other structures.

The ramification of this modular structure is that variants that arise on this backdrop have the potential of also acting modularly, the main benefit of which from an evolutionary angle is the site-specific sculpting of a structure towards an adaptive goal. However, such variants, which may also confer deleterious effects (e.g., late-life disease risk), are likely to be more tolerated due to the specificity of their regulatory effects, especially compared to those that reside in more pleiotropic regulatory (and coding) sequences which should be under stronger purifying selection for their negative fitness impacts. For example, we previously found that there are regulatory elements unique to the distal femur and unique to the proximal tibia of the human embryonic knee that show evidence of past positive selection in the hominin lineage. In contrast, regulatory elements shared between these bone-ends did not display ancient selection signals[20]. Interestingly, distal femur- and proximal tibia-specific regulatory elements were also observed to be enriched in GWAS OA risk variants[20]. Overall, these data suggest that at GWAS loci exhibiting multiple diseases/traits associations, distinct variants may underlie each association or sets of related associations because of this evolution-driven modularity, and the need to balance positive and negative selection. In other words, both at the regulatory element level as well as the sequence level, the evolution of a specific locus (in the context of its form and function) likely shapes disease risk and trait variation because of the interplay between modularity, pleiotropy, and evolution.

We do acknowledge several caveats that limit our findings and interpretations, namely that not all of the musculoskeletal traits/

diseases used in our studies are solely chondrocyte mediated, and thus the genome-wide modularity patterns we observed in chondrocyte regions may not hold in all cases. However, at this time, similar datasets do not exist for osteoblasts or other skeletal cell types at site-specific levels. We were also limited in our numbers of anatomical sites to study chondrocyte regulation, as well as constrained by the enormity of the task to find and connect all relevant variants at a given locus to a regulatory sequence and then to a gene.

Finally, our findings point to an important practical issue, namely that while we observed modularity at the level of accessibility (at *GDF5* for example), much of the functional consequences of this modularity became apparent upon modeling each variant in a proper three-dimensional in vivo system. We therefore, need better systems to more rapidly model and phenotype variants in vivo in order to detect their functional effects on aspects of three-dimensional anatomy and over time, which is critical to many diseases, not just those of the musculoskeletal system. Overall, we advocate for a multifaceted framework to study the genetic architecture of complex traits/diseases, one which combines high throughput functional genomic screens on relevant rare developmental human and patient tissues, locus-specific functional modeling of enhancers and risk variants within human cells and in the mouse model (when appropriate), and the use of relevant preclinical and clinical human datasets (yielding both genetic and phenotypic data). These methods should help decode the role of regulatory variants in disease mechanisms, as well as those involved in adaptive evolution.

## Methods

**ATAC-seq data collection and analysis**. Raw sequencing FASTQ files and processed peak bed files have been deposited on NCBI GEO (GSE153260).

Mouse samples: Mouse ATAC-seq data were acquired on gestational day (E) 15.5 mouse tissues (proximal and distal femur and tibia) from Richard et al.[20] with methods described (Supplementary Data 2). At this developmental time point, chondrocytes are easily extracted from the extracellular matrix for ATAC-seq with little impact on epigenetic state[56]. Briefly, embryonic cartilaginous ends from the femur and tibia were dissected under a microscope in 1× phosphate-buffered saline (PBS) on ice and soft tissues were removed. Each proximal or distal chondrocyte population was micro-dissected from bony diaphyses and separately pooled from a single litter, consisting on average of eight animals (*n* = 2 or 3 per tissue)[20]. All samples were processed using our optimized ATAC-seq protocol as described below and previously[20,56]. All processed chromatin samples were subjected to polymerase chain reaction (PCR) amplification, barcoding, and post amplification processing following methods[20,55,56]. Samples were then sent to the Harvard University Bauer Core Facility for sequencing on the Illumina NextSeq500. Sequencing yielded ~400 million reads per lane and an average of 50 million per sample. See our previous publication[20,56] for summary statistics and all relevant sequencing files.

Human samples: Human products of conception at E67 (*n* = 3) were collected from the first-trimester termination through the Laboratory of Developmental Biology at the University of Washington in Seattle, the University of Washington in full compliance with the ethical guidelines of the National Institutes of Health (NIH) and with the approval of the University of Washington Institutional Review Boards for the collection and distribution of human tissues for research, and Harvard University for the receipt and use of such materials. The Laboratory of Developmental Biology obtained written consent from all tissue donors. The University of Washington Birth Defects Research Laboratory was supported by NIH award number 5R24HD000836 from the Eunice Kennedy Shriver National Institute of Child Health and Human Development (NICHD). Harvard University IRB determined this sample constitutes Non-Human Subjects Determination Status (Capellini: IRB16-1504). PI Capellini received no federal funds (e.g., NIH) to acquire, receive, process, or utilize these samples. The human samples were briefly washed in Hank's Balanced Salt Solution (HBSS) and transported at 4 °C during shipment. Upon arrival the samples were dissected under a light dissection microscope in an identical fashion to all mouse samples reported above and subjected to the ATAC-seq protocol described in Richard et al.[1] and following approved Harvard University IRB (Capellini: IRB16-1504) and COMS (Capellini: 18-103) protocols. For this study, all hind limb elements (including pelves) were intact but then separated from one another manually, allowing to study each cartilaginous structure in isolation from bony diaphyses. Samples were then processed accordingly using the ATAC-seq pipeline as performed for mouse samples. Libraries for proximal and distal bone end samples for each element (acetabulum of pelvis, proximal and distal femur and proximal and distal tibia),

were then sent out for sequencing running a pooled library for three rounds of sequencing on the Illumina NextSeq 500 in order to obtain at least 50 million reads per sample, one sample (acetabulum from biological replicate #2) reached this number of reads in two lanes, so the third lane of sequencing was not run. Quality control statistics and primer information are presented in Supplementary Data 6.

**ATAC-seq read processing**. Processing of all ATAC-seq datasets was conducted using computational pipelines outlined in detail in Richard et al.[20] and below. For mouse samples, the mm10 coordinates of Irreproducible Discovery Rate (IDR) called peaks per each tissue were acquired from Richard et al.[20] and Guo et al.[56] and are presented in Supplementary Data 2. For human samples, all sequencing files were quality checked with FastQC. Sequenced reads across runs were pooled for each sample and adapters were removed with NGMerge[82]. Reads were then aligned to the human reference hg19 genome assembly with Bowtie2 v2.3.2[83] using default parameters for paired-end alignment. Reads were filtered for duplicates using Picard's MarkDuplicates and mitochondrial reads were also removed. BAM files were subsequently used for peak calling using MACS2[84] software (version 2.1.1.2), using the following flags for 'callpeak':–BAMPE–nolambda. For human samples, peaks (referred to as "regulatory elements") reproducible across biological replicates were screened using an IDR threshold of <0.05, as defined by the IDR statistical test[85] (version 2.0.3) (Supplementary Data 2). For comparisons between mouse and human ATAC-seq data called peaks were lifted-over from mm10 to hg19 using the UCSC 'liftover' utility[86] using the following flags: '-minMatch=0.1 -bedPlus=4', with the relevant liftover chain file similarly obtained from UCSC.

**ATAC-seq peak/GWAS variant intersections**. Please see the section entitled "ATAC-seq peak and risk variant intersection to whittle-down the *GDF5-UQCC1* knee OA and DDH association intervals" in Supplementary Information. GWAS variant data on knee OA and DDH are presented in Supplementary Data 3.

**Examination of modularity genome-wide and at GWAS loci**. All Refseq gene promoter TSS sites for the hg19 genome were obtained from the UCSC Table Browser[87], with windows defined as 100 kb up/downstream of each site. Called IDR peaks for all tissues (as presented in Supplementary Data 2) were subsequently intersected with these sites using bedtools version 2.29.2[88]; peaks overlapping between different tissues were merged using the bedtools 'merge' function. For each of these per-gene windows we considered the behavior of intersected peaks. For each peak, we considered the tissues for which overlaps were found (e.g., a peak overlapping in the proximal and distal femur). Across all (merged) peaks around a gene, if the tissue representation of each peak was identical (i.e., all peaks around a gene are overlapped in the proximal and distal femur) then the gene was considered to be 'non modular'. Conversely, gene windows for which at least two different tissue-overlap patterns of nearby peaks were observed were considered to be 'modular' (i.e., have some modularity in putative cis-regulation). We also classified individual peaks as having some degree of modularity (i.e., were not represented in all tissues), versus those which were overlapped with peak calls from all tissues (i.e., are 'pleiotropic' putative enhancer elements), and those for which peak calls were only observed in one tissue (i.e., are 'singletons'). Gene windows only intersecting 'pleiotropic' putative enhancer elements were themselves considered to be 'pleiotropic' with regards to local accessibility.

For each gene window defined above, we next intersected the set of musculoskeletal GWAS variants we had aggregated (see Supplemental Information, Supplementary Data 1), after lifting over from hg38 to hg19 using the 'liftOver' utility[86]. For each individual gene window, the set of intersected SNPs were collapsed and the number of different GWAS traits represented in a given window was counted—those windows for which multiple different trait associations were intersected were considered 'recurrent risk loci'. The lists of genes defined as 'modular' and 'recurrent risk loci' were then compared using the 'phyper' function in R version 4.0.2[89,90]. This was done comparing the number of 'recurrent risk loci' genes which were also considered 'modular' to the number of 'recurrent risk loci' not also considered to be 'modular'. In addition, the number of 'modular' genes which were also considered risk loci (generally, not necessarily recurrent) was compared to the number of 'modular' genes which were not considered risk loci using a similar hypergeometric test.

**DDH patient sample collection, processing, and measurements**. DDH patient sample collection: The study was approved by the human ethics committee and Institutional Review Board of the Affiliated Zhongshan Hospital of Dalian University, China. A total of 120 femur heads from patients with DDH were collected, albeit only 113 were further studied given genotyping issues with seven samples (see below). All femoral heads from patients undergoing total hip arthroplasty due to DDH were obtained with signed consent. The diagnosis of hip dysplasia was made by conventional radiographs (X-Ray) and/or computed tomography (CT) with a center-edge angle of Wiberg of <20° measured on a well-centered antero-posterior radiograph of the pelvis. The affected areas of articular cartilage from the femoral heads were cut into pieces (1 × 1 × 1 cm$^3$) immediately after they were collected. Half of the specimens were fixed in 4% paraformaldehyde for histology and the other half were snap-frozen in liquid nitrogen within 30 min of explanation and stored at −80 °C.

DDH patient sample genotyping: SNP genotyping of rs4911178 was performed using the Sanger sequencing method. Briefly, total genomic DNA (gDNA) was extracted from archive paraffin-embedded or fresh tissue samples collected from DDH patients using the Allprep DNA/RNA FFPE kit (Qiagen, Hilden, Germany) following the manufacturer's protocol. PCR was performed using 100 ng template gDNA, rs4911178 Forward1 and rs4911178 Reverse1 primers (Supplementary Data 6) with annealing temperature at 55 °C, 200 µM ddNTPs, 1.5 mM magnesium chloride, and HotStarTaq DNA Polymerase (Qiagen, Hilden, Germany). The PCR products were purified and sequenced by Sangon Biotech (Shanghai, China) using the ABI 3730XL DNA Sequencer (Applied Biosystems). Of the 120 samples, seven samples failed in the acquisition of usable DNA for accurate genotypes, and these seven samples were removed from all downstream analyses. These data are presented in Supplementary Data 4.

Clinical DDH patient morphometric analyses: Acetabulum inclination angle of sharp, acetabular index (Tönnis angle), and center-edge angle were measured from preoperative X-rays[91,92]. These measurements were compared between the hips with homozygous (A/A) and heterozygous (A/G) risk variants. These comparisons are presented in Fig. 2a. Among all the hip measurements examined to assess DDH, Sharp angle is the only measurement that can be also replicated in mice. The measured inclination angle was then compared between different patient genotypes as well as to published normative data in Asian subjects with normal hips[91,93]. These data are presented in Supplementary Fig. 3.

**Knee OAI patient sample processing and measurements.** Knee OAI database samples: The OAI dataset consisted of 4,129 individuals identified as either being at risk of developing, or suffering from, OA. This set contains groups of 3,366 Caucasian and 763 Black/African American individuals (on the basis of self-reported ethnicity); in order to control for potential effects of demographic history, only Caucasian individuals were considered for genetic analyses (see below). This group ranged in age from 45 to 79 (average of 62), BMI from 17.6 to 46.8 (average of 28), consisting of 1,498 males and 1,868 females (Supplementary Data 5).

**Knee OAI genetic analyses.** Genotyping data for study participants in the OAI dataset was obtained through the database of Genotypes and Phenotypes (dbGaP) with appropriate permissions ($n = 4,129$); self-identified race was used to separate the dataset into White/Caucasian ($n = 3,366$) and Black/African-American ($n = 763$) groups. Given the disparate sampling sizes of these two groups, and to avoid potential demographic signals on genotype, only individuals in the White/Caucasian group were further analyzed. Genotyped SNPs were extracted using PLINK version 1.9[94]. hg18 genotyping files were then lifted over to hg19 using CrossMap version 0.3.3[95] using chain files obtained from UCSC. BEAGLE5[96] was used to impute variants from the 1KG3 European reference panel using the 'conform-gt.jar' and 'beagle.28Sep18.793.jar' utilities, leaving all settings to default. Subsequently, genotype information on rs6060369 (imputed) was extracted for White/Caucasian individuals.

To look at the prevalence of the 'T' risk allele at rs6060369 in different subsets of the OAI population, we first separated patients into those who presented with no/moderate-OA (KL = 0/1) ($n = 1,207$) in either knee at baseline and all subsequent follow-ups, those with significant OA (KL ≥ 2) in either knee at baseline ($n = 1,119$), and those who presented with no/moderate-OA, but progressed to significant OA in either knee at the time of the last follow-up in the study ($n = 208$) (Supplementary Data 5, Sheet 2). In order to compare the frequency of the 'T' risk allele at rs6060369 between these groups, sub-sampling to $n = 200$ (approximately the size of the smallest group) was done, with replacement, 200 times. Sub-sampling at additional N-values was done ($n = 50, 100$), with calculated differences/significance values similar to those observed using $n = 200$. The number of 'T' alleles was summed over all individuals in a sub-sample, resulting in 200 data points per group. Sub-sample allele counts were compared between groups using a two-tailed Student's T-test and adjusted for $n = 3$ comparisons. In order to ensure the robustness of comparison tests to subsample-averaging, first and third quartile values for risk-allele counts in subsamples were also compared. To confirm that comparison results were not sensitive to spurious randomized sampling effects, the above algorithm was applied 1000 times, with the average adjusted p-value and t-statistic from repeat tests calculated. For graphical purposes, the average group-subsampled allele counts for each of 1000 iterations were taken and used to generate the density plots shown in Supplementary Fig. 9. The above group-comparison methodology was also used in a simplified group comparison between individuals presenting with ($n = 1,868$) or without ($n = 1,393$) significant OA in either knee. (Supplementary Data 5, Sheet 1).

Knee OAI morphometric analyses: Baseline MRI images of OAI patients with KL = 0 at the time of enrollment ($n = 383$) were used to quantify bicondylar width of the femur, condylar width (medial and lateral), condylar curvature (medial and lateral), notch width, width of the tibial plateau, posterior tibial slope (medial and lateral), maximum depth of the medial tibial plateau, tibial, tibial spine height (medial and lateral), cross-sectional area of the posterior meniscus (medial and lateral), tip angle of the posterior meniscus O (medial and lateral), anterior cruciate ligament (ACL) length and sagittal elevation angle. The measurements were done based on established clinically relevant protocols and showed strong inter- and intra-examiner reliability (ICC > 0.8).

**Animal models.** The following five mouse lines were used in the study:

(1) The $R4^{+/-}$ enhancer null mouse line contains a deletion of the $R4$ enhancer and was generated on the C57BL/6J *Mus musculus* background by PI Capellini laboratory. This line is described in Richard et al.[20].

(2) The $R4^{rs6060369-T/rs6060369-+}$ single allelic replacement mouse line contains a single "T" allelic base-pair replacement of the orthologous human rs6060369 variant in the $R4$ enhancer and was generated on the C57BL/6 J *Mus musculus* background by Applied StemCell and PI Capellini laboratory. This line is described in detail in Richard et al.[20].

(3) The *GROW1* enhancer *LacZ* line contains the human *GROW1* enhancer cloned upstream of a minimal promoter and the *LacZ* reporter gene and was generated via transgenesis on the FVB/NJ *Mus musculus* background by the Harvard Genome Modification Facility and the Capellini laboratory. Upon receipt of founder mice, the line was continually backcrossed on the FVB/NJ background for at least four generations. We followed standard breeding and husbandry protocols to maintain and expand the line; and both males and females were used in experimental studies examining expression at joint sites at embryonic timepoints.

(4) The $GROW1^{+/-}$ enhancer null mouse line contains a deletion of the $GROW1$ enhancer and was generated on the C57BL/6 J *Mus musculus* background by the PI Capellini Laboratory. This line is described in detail in Capellini et al.[48].

(5) The $GROW1^{rs4911178-A/rs4911178-+}$ single allelic replacement mouse line contains a single "A" allelic base-pair replacement of the orthologous human rs4911178 variant in the $GROW1$ enhancer and was generated on the C57BL/6 J *Mus musculus* background by Applied StemCell, Inc. and the Capellini laboratory. Upon receipt of founder mice, the line was continually backcrossed on C57BL/6 J for at least four generations. See specific comments below in the section entitled "Generation of $GROW1^{rs4911178-A/rs4911178-+}$ mice". We followed standard breeding and husbandry protocols to maintain and expand the line; and both males and females were used in post-natal experimental studies examining morphology and histology at P56, with cohorts of different genotypes per sex being subjected to the above-mentioned methods.

For all mouse lines, all breeding, husbandry, euthanasia, and experimental protocols strictly followed IACUC-approved protocols (Capellini: 13-04-161-2) at Harvard University.

**Generation of $GROW1^{rs4911178-A/rs4911178-+}$ mice.** Targeting of the mouse orthologue of the human rs4911178 variant for replacement with the human DDH risk variant at this locus was performed in collaboration with Applied StemCell, Inc. As reported in Capellini et al.[48], all sequenced mammals possess a "G" variant at this position, a finding that is also true for all MGI mouse strains, and all African apes assessed from the Great Ape Genome Diversity Project. Thus, currently, the "A" variant is unique to humans. CRISPR-Cas9 gene targeting was used to introduce a single base-pair change at (mm10) chr2:155,899,861 in the C57BL/6J mouse strain, in which the ancestral "G" variant was substituted by the human DDH/OA risk "A" variant[48]. To achieve this, a mixture containing in vitro transcribed active sgRNAs (Supplementary Data 6), a single-stranded oligodeoxynucleotide, and Cas9 protein was first microinjected into C57BL/6 J embryos acquired from Jackson Laboratory, Bar Harbor, ME. Next, initial founder mice were screened for variant changes by extracting genomic DNA from tail tissues and then PCR amplifying the target region using Chr2-Reg-I-PM Forward1 and Reverse1 primers (Supplementary Data 6) and the following conditions: All PCR amplifications were prepared in 25 µL using MyTaqTM Red Mix (Bioline, Cat#, BIO-25044) and the amplifications were carried out using the following program: 95 °C, 2 min; 35 cycles of [95 °C, 15 s; 60 °C, 15 s; 72 °C, elongation duration varies by amplicon size], 72 °C, 5 min; 4 °C, indefinite. Finally, Next-generation sequencing (NGS) libraries were prepared and subsequently sequenced using Illumina PCR genotyping to identify G > A point mutation at Chr2:155,899,861.

After microinjecting >300 embryos, and screening of three potential founders, we recovered two F0 mice with the "A" replacement confirmed using NGS on DNA extracted from their tail tissues. As assessed by sequencing, both F0 founders lacked any other artifactual alteration to the locus. These F0 mice were then crossed once again to wild type C57BL/6J mice to confirm allelic transmission, upon which we recovered two separate transmitting F1 C57BL/6J $GROW1^{rs4911178-A/rs4911178-G}$ lines (Line 1 and Line 2). Both lines were then backcrossed for four generations to C57BL/6J wild type mice to remove possible off-target effects produced by the CRISPR-cas9 process. In addition, given the finding that sgRNAs can cause local alterations (deletions/insertions/rearrangements/substitutions) within a 5 kb vicinity of the intended target site[97], homozygous replacement mice were screened using local primers targeting this interval followed by Sanger sequencing, resulting in no non-intended modifications. Subsequently, $GROW1^{rs4911178-A/rs4911178-G}$ mice were intercrossed to generate $GROW1^{rs4911178-G/rs4911178-G}$, $GROW1^{rs4911178-A/rs4911178-G}$, and $GROW1^{rs4911178-A/rs4911178-A}$ mice for all downstream phenotyping experiments. Similar crosses were previously conducted for $R4^{rs6060369-T/rs6060369-+}$ mice, see below (Richard et al.[20]). $GROW1$ mice were also crossed to wild type 129×1/SvJ mice for ASE studies (see below).

**Expansion of $R4^{rs6060369-T/rs6060369-+}$ mice for morphometric and histological study**. To assess the OA and cartilage status of $R4^{rs6060369-+/rs6060369-+}$ and $R4^{rs6060369-T/rs6060369-T}$ mice, right hind limbs of 15 male 1-year-old mice (5 wild type and 10 homozygous) were obtained for histological analysis (Supplementary Table 2). After skin and excess muscle removal, each limb was fixed in 10% neutral buffered formalin for 24 hours. Each limb was next decalcified in 14% Ethylenediaminetetraacetic acid (EDTA) at pH 7.5 for 8 days, and then dehydrated by incubation in 70% ethanol, 95% ethanol, 100% ethanol, 1:1 ethanol-xylene-solution, 100% xylene (all at room temperature), and paraffin (at 60 °C) (Sigma). The incubation period was 1 day for these specimens, after which they were bisected in a frontal plane and embedded. The formalin-fixed paraffin-embedded tissues were then sent to the Massachusetts General Hospital Center for Skeletal Research (CSR) Histology & Histomorphometry Services core facility and processed in one batch for proper sectioning and histological staining using H&E and Safranin O staining. Embedding, sectioning, and staining were performed without knowledge of genotype.

Upon receipt of sections, a panel of two readers (AMK and DM), who were blinded to genotype, scored sections for knee OA by consensus. The scoring was based on the Osteoarthritis Research Society International (OARSI) histopathology initiative recommendations for the histological assessment of OA in the mouse. Briefly, medial tibial plateau (MTP), medial femoral condyle (MFC), lateral tibial plateau (LTP), and lateral femoral condyle (LFC) were assigned a semiquantitative score ranging from 0 to 6, with 0 representing a normal articular cartilage and 6 representing vertical clefts/erosion to the calcified cartilage extending >75% of the articular surface. The sum score of all four joint surfaces was determined (referred to as OARSI Score in the manuscript) and selected as primary outcome. A Mann–Whitney test was used to compare the OARSI scores between the wild type ($n = 5$) and homozygous ($n = 10$) knees. $P$ values are two-sided and the statistical significance was assessed at alpha = 0.05.

**Cell lines and culture conditions**. T/C-28a2 human chondrocyte cells (female) were acquired from Dr. Li Zeng (Tufts University) courtesy of Dr. Mary Goldring (The Hospital for Special Surgery). This line was cultured at 5% $CO_2$ at 37 °C in Dulbecco's Modified Eagle's Medium (DMEM), 10% fetal bovine serum (FBS), and 1% penicillin-streptomycin (P/S). Media was replaced every 2–3 days and the cells were subcultured every 5 days. After receipt of the cell line from the source institution, cells were passaged and used in experimental assays without additional STR authentication or mycoplasma testing.

**Micro-CT and anatomical measurements in mice**. To quantify phenotypes in developed mouse models (Supplementary Table 2), the right pelvis, femur, and tibia were harvested and imaged using high-resolution Micro-Computed Tomography (µCT40, SCANCO Medical AG, Brüttisellen, Switzerland). Scan parameters were: 12 µm³ isotropic voxel size, 70 kVp peak X-ray tube intensity, 114 mA X-ray tube current, and 200 ms integration time. Digital Imaging and Communications in Medicine (DICOM) images were exported for measurements of the following key anatomical features in Osirix MD v7.5 (Pixemo SARL, Bernex, Switzerland)[20,48,62,63].

Pelvis: acetabular depth, acetabular diameter, acetabular inclination, acetabular anteversion.
Proximal Femur: valgus cut angle, neck-shaft angle, neck length, neck diameter, head offset, head diameter.
Distal Femur: bicondylar width, notch width, condylar width (medial and lateral), condylar curvature (medial and lateral), trochlear width (medial, central, and lateral), trochlear groove depth, trochlear sulcus angle.
Tibia: tibial plateau width, posterior tibial slope (medial and lateral), tibial spine height (medial and lateral).

The measurements were done based on established clinically relevant protocols and showed a strong inter- and intra-examiner reliability (ICC > 0.78).

Micro CT images were also used to generate 3D models of each bone using the bone segmentation process in a commercially available image processing software (Mimics v17.0, Materialise). The 3D models were then imported to 3-matic software package (v9.0, Materialise) and then co-registered together using a global n-point registration technique. The registered models from the wild type and homozygous mice were then used to generate 3D heatmaps indicating the geometrical differences between wild type and homozygous mice for each line. The heatmaps were generated by calculating the distance between the corresponding points in co-registered models, where dark blue indicates the maximum deviation in the negative direction and red indicates the maximum deviation in the positive direction.

**X-gal staining**. Whole-mount staining for β-galactosidase activity was performed as below and as described[20,48,61]. GROW1 enhancer LacZ positive embryos were fixed in 4% paraformaldehyde (PFA) (Sigma, 158127) at 4 °C, according to gestational-day guidelines. Fixed embryos were washed three times in LacZ wash buffer and stained for 16 h in the dark with 1 mg/ml X-gal (5-bromo-4-chloro- 3-indolyl-β-D-galactopyranoside) (Sigma, B4252) in staining buffer at room temperature. After staining, embryos were briefly rinsed in the wash buffer, post-fixed in 4% PFA at 4 °C for 5 h and stored in 1× PBS.

**ASE analyses**. Timed mating was established between C57BL/6J $GROW1^{rs4911178-A/rs4911178-G}$ heterozygous mice and 129×1/SvJ $GROW1^{rs4911178-G/rs4911178-G}$ wild type mice. Previous studies on ASE assays on $GROW1^{+/-}$, $R4^{+/-}$, and $R4^{rs6060369-T/rs6060369-+}$ can be found at Capellini et al.[19] and Richard et al.[20], but follow the same procedures provided below. Pregnant females were sacrificed according to IACUC-approved protocols to acquire E15.5 embryos. Right and left hind limbs were stripped of all soft tissues, and proximal and distal femoral and proximal tibia chondrogenic tissues were dissected from each limb, with each tissue (e.g., left and right distal femur) placed in TRIzol reagent (15596-026, Ambion by Life Technologies) with a homogenizer bead, and then homogenized for 2 min with a tissue homogenizer (Qiagen). Samples were then stored at −80 °C. RNA was then isolated with TRIzol reagent using Direct-zol™ RNA Miniprep Kit (supplied with DNase I, Zymo). For GROW1 rs4911178 mouse orthologue variant tests: $n = 7$ proximal femur replicates and $n = 7$ distal femur replicates. Samples were then run on a Bioanalyzer to ensure RNA integrity numbers >8. These RNA samples were then reverse transcribed using SuperScript III First-Strand cDNA Synthesis Reaction kit (18090010, Life Technologies) according to the manufacturer's recommendations. Independently, tails from each embryo were used for genotyping as described above.

cDNA samples were then sent to EpigenDx for ASE-assay design and execution. SNPs in the coding regions of Gdf5 (rs27340038), Cep250 (rs27339949), and Uqcc1 (rs3684985) were identified by EpigenDx. Pyrosequencing for SNP genotyping (PSQ H96A, Qiagen Pyrosequencing) is a real-time sequencing-based DNA analysis that quantitatively determines the genotypes of single or multiple mutations in a single reaction. Briefly, 1 ng of cDNA sample was used for PCR amplification. PCR was performed with 10X PCR buffer (Qiagen) with 3.0 mM $MgCl_2$, 200 M of each dNTP, 0.2 µM each of the forward and reverse primers (available through EpigenDx), and 0.75 U of HotStar DNA polymerase (Qiagen) per 30 µl reaction. The PCR cycling conditions were 94 °C for 15 min; 45 cycles of 94 °C for 30 s, 60 °C for 30 s, 72 °C for 30 s; and 72 °C for 5 min. One of the PCR-primer pairs was biotinylated to convert the PCR product to single-stranded DNA-sequencing templates with streptavidin beads and the PyroMark Q96 Vacuum Workstation. 10 µl of the PCR products were bound to streptavidin beads, and the single strand containing the biotinylated primer was isolated and combined with a specific sequencing primer (available through EpigenDx). The primed single-stranded DNA was sequenced with a Pyrosequencing PSQ96 HS System (Qiagen Pyrosequencing) according to the manufacturer's instructions (Qiagen Pyrosequencing). The genotypes of each sample were analyzed with Q96 software AQ module (Qiagen Pyrosequencing).

Pyrosequencing results for each SNP were used to calculate the allelic ratios of, for example, C57BL/6J GROW1 rs4911178 mouse orthologue "A" allele in the heterozygous state) to 129×1/SvJ (wild type "G" allele). Each ratio of cDNA products found in heterozygous animals was then normalized by the ratio of wild type C57BL/6J to 129×1/SvJ genomic products, amplified from known 1:1 mixtures of each sequence. The permutation test, a non-parametric measure, was used to determine significance between the wild type and heterozygous allelic expression ratio with the[98] module in R.

**CRISPR targeting of GROW1 and rs4911178 in vitro**. All sgRNAs flanking (1) the human GROW1 enhancer and (2) a 13 bp region within the human GROW1 enhancer containing the DDH/OA risk variant rs4911178 were designed using MIT CRISPR Tools (http://crispr.mit.edu), synthesized by Integrated DNA Technologies, Inc (Coralville, Iowa), and cloned into the PX458 vector following published protocols[13]. The sequence of all sgRNAs along with their chromosomal locations (hg19) are listed in Supplementary Data 6.

Guide RNAs, flanking the GROW1 element, and guide RNAs flanking a 13-bp region containing DDH/OA risk variant rs4911178 were first tested for the ability to induce efficient deletions of the human element in cultured T/C-28a2 chondrocytes ($n = 3$ biological replicates per assay). T/C-28a2 cells were maintained in DMEM (Gibco, Gaithersburg, Maryland) supplied with 10% FBS (Gibco) and 1% Pen/Strep (0.025%) and seeded in a six-well plate for 1- day prior to transfection. After culturing at 37 °C with 5% $CO_2$, we scanned cells under a GFP-microscope to verify successful GFP transfection efficiency (i.e., >70% of the cells were GFP positive). DNA was then extracted from the cells using E.Z.N.A Tissue DNA Kit (Omega Bio-Tek, Norcross, GA), and the GROW1 regulatory element was amplified using PCR primers flanking each sgRNA location (listed in Supplementary Data 6). PCR amplified products were purified from 1% agarose gel (E.Z.N.A Gel Extraction Kit). Sanger sequencing was used to verify successful targeting of the larger GROW1 region, whereas amplicon sequencing at the MGH Center for Computational and Integrative Biology (CCIB) DNA core was used to verify successful targeting of the smaller 13-bp GROW1 regulatory region containing rs4911178. To examine effects on GDF5 and nearby gene expression, RNA was extracted from control and CRISPR-Cas9 targeted T/C-28a2 cells ($n = 3$ biological replicates, with three technical replicates per experiment per condition) and prepared using Trizol Reagent (Thermo Fisher Scientific, Springfield

Township, New Jersey) and Direct-zol™ RNA Miniprep kit (ZYMO). Two micrograms of total RNA were used to synthesize the first-strand cDNA using SuperScript III First-Strand Synthesis System (Thermo Fisher Scientific). qRT-PCR analysis was then performed with specific primers per gene and Applied Biosystems Power SYBR master mix (Thermo Fisher Scientific) with *GAPDH* house-keeping gene as an internal control. Primers used for qRT-PCR are listed in Supplementary Data 6. CRISPR-Cas9 experiments performed for *R4* and rs6060369in T/C-28a2 cells are described in Richard et al.[20].

**ChIP assay**. ChIP on human chondrocyte cell line: PITX1 ChIP assay on human chondrocyte cell line T/C-28a2 was performed as below and as previously described[20]. The cell line was chosen to maximize the number of cells needed to perform transcription factor ChIP. The strand-specific PCR primers used for amplifying the 234 bp region containing rs4911178 base-position in the human genome are indicated in Supplementary Data 6.

ChIP assay on *GROW1* rs4911178 replacement mice: Timed mating's were performed between homozygous C57BL/6J *GROW1*^rs4911178-A/rs4911178-A^ and wild type *GROW1*^rs4911178-G/rs4911178-G^ mice. Pregnant females were sacrificed according to IACUC-approved protocols. Embryonic hind limb buds (proximal femoral, distal femoral, and proximal tibia) at E15.5 were collected separately into a microcentrifuge tube containing 200 μL of 5% FBS supplemented DMEM medium. The collected tissues were subjected to 1% collagenase, type 2 (Cat. No. LS004176, Worthington Biochemical Corp. NJ, USA) digestion for 2 h at 37 °C rocking, gentle mixing every 30 minutes to generate single-cell chondrocyte suspension. The dissociated single cells were then washed twice with 1× PBS and subsequently, ChIP-assay was performed on these cells using the methods described in Richard et al.[20]. Three biological replicate ChIP-assays (n = 3) were performed on each tissue. The strand-specific PCR primers used for amplifying the 255 bp region containing the orthologous replacement rs4911178 base-position in the mouse genome are indicated in Supplementary Data 6. PCR amplification was performed in a 50 μl reaction mixture containing 2 μl of input/ChIP DNA by the addition of 0.02 U/μl Q5® High-Fidelity DNA Polymerase. A hot start was performed (98 °C for 30 s), followed by 35 cycles at 98 °C for 10 s, 58 °C for 30 s, and 72 °C for 30 s with a final extension at 72 °C for 2 min. The PCR product was separated on 1 % agarose gel containing 0.05 μg/mL of ethidium bromide (EtBr) and photographed using VWR photoimager Dual UV transilluminator system (VWR International, USA).

For NGS of amplicons (Amp-Seq) for *GROW1* rs4911178 sequences, a 255 bp segment harboring the orthologous rs4911178 position was PCR amplified from input control and ChIP-DNA. The amplicons were purified, quantified using Nanodrop, and sent to the MGH DNA core facility for CRISPR amplicon sequencing service. A summarization analysis was performed using CRISPResso, a web-based tool used for the analysis of genome editing outcomes from deep sequencing data[99]. CRISPResso tool reported sequences, and their abundances, present in the sequenced samples. Following the acquisition and analysis of NGS data for each of the three ChIP assay biological replicates per tissue, for each assay the allelic ratio of, for example, the risk "A" to non-risk "G" (i.e., A:G) at rs4911178 in the *GROW1* enhancer was calculated first from the ChIP-input generated NGS results, and then from the ChIP PITX1 pull-down generated NGS results. After, for each assay, the ratio of A:G from the ChIP pull-down was normalized by the ratio of A:G from the ChIP Input, which permitted an assessment of the gain in "A" allelic binding by PITX1. A student's *T*-test was used to compare this ratio to the expected A:G ratio in the ChIP input.

**PITX1 over-expression in mouse chondrocytes**. Timed mating's were established between homozygous C57BL/6J *GROW1*^rs4911178-A/rs4911178-A^ mice and 129×1/SvJ *GROW1*^rs4911178-G/rs4911178-G^ wild type mice. Pregnant females were sacrificed according to IACUC-approved protocols to acquire E15.5 embryos. Right and left hind limbs were stripped of all soft tissues, and proximal femoral, distal femoral, and proximal tibia chondrogenic tissues were dissected from each limb and subjected to 1% collagenase II treatment for 2 h at 37 °C rocking, mixing every 30 min. The samples were then filtered using 70-μM nylon cell strainer and the filtered single cells were cultured in DMEM medium supplemented with 10 % (v/v) FBS and 1 % (v/v) Pen/Strep (0.025%) at 37 °C in humidified 5% CO2 incubator. Proximal/distal femoral and proximal tibial-derived chondrocytes were transfected with PITX1-overexpression vector, pCMV-SPORT6-PITX1 (Dharmacon, Inc. Chicago, IL) at 50-80% confluence in 12-well plates using Lipofectamine 2000 transfection reagent (Life Technologies). For T/C-28a2 cell transfection, vector concentrations of 50, 100, 500, and 1000 ng were initially tested for efficient PITX1 overexpression. 100 ng showed optimal PITX1 overexpression and was used for all T/C-28a2 experiments. Cells were then transfected transiently with 100 ng/well of PITX1-overexpression vector. The total amount of DNA transfected in each well was kept constant. All transfection analyses were performed in triplicate. Cells were harvested ~48 h following transfection and the total RNA was extracted from the transfected cells using TRIzol reagent and Direct-zol™ RNA Miniprep Plus Kit (supplied with DNase I, Zymo). Reverse transcription was performed using SuperScript III First-Strand Synthesis System (Thermo Fisher Scientific). qRT-PCR was performed for *PITX1*, *Gdf5*, and *Pgk1*, a housekeeping gene using primers listed in Supplementary Data 6. The fold-change in expression for *PITX1* and *Gdf5*

genes were calculated with *Pgk1* as an internal control. PITX1 ChIP assay was performed on primary femur chondrocytes over-expressed with PITX1. The input and ChIP DNA amplicons were purified and sent to the MGH DNA core facility for CRISPR amplicon sequencing service. For each assay the allelic ratio of (i.e., A: G) at rs4911178 in the *GROW1* enhancer was calculated using the method described above.

### Quantification and statistical analyses

*Mouse and human morphometric analyses.* For mouse morphometric data, each anatomical feature was defined as a continuous variable and then compared between the genotypes (wild type, heterozygous and homozygous) using a one-way analysis of variance (ANOVA). Dunnet posthoc was used for pairwise comparisons between heterozygous and wild type and homozygous and wild type. Similar approach was used for knee morphology assessments in OAI patients. Two-sample *t*-test was used to compare the acetabular inclination angle of Sharp, acetabular index (Tonnis angle), and center-edge angle in the DDH cohort between the homozygous (n = 83) and heterozygous (n = 25) genotypes. Linear regression analysis was used to assess the associations between reductions in *Gdf5* expression and changes in anatomy. The regression coefficient (β) was used to find the relative effect size of the reduction on *Gdf5* expression on changes in anatomy. The ratio of average changes in anatomy in homozygous vs heterozygous mutations was calculated to find the relative influence of homozygosity on joint morphology. Measurements were done by an experienced reader blinded to genotypes. To assess measurement reliability, a randomly selected subset of images (20 in mouse models and 20 in human cohorts) were reanalyzed by the same examiner and two additional examiners. The inter- and intra-class correlation coefficients for each anatomical feature were calculated. Analysis was conducted in SPSS (v27, IBM Corp., Armonk, NY). *p*-values are two-sided, and the statistical significance was assessed at alpha = 0.05.

Human knee OAI genetic analyses: Individuals in the OAI study, separated by KL grade, were randomly sub-sampled with replacement to match the size of the smallest set (n = 200, and also n = 50,100) for 200 sub-sample iterations. Rs6060369 'T' alleles were counted per subset, with sets of allele-counts per group compared using a two-tailed Student's *T*-test, adjusting significance for n = 3 comparisons. For robusticity across random sampling, this subsampling/testing was performed 1000 times, with the average/standard-deviation *t*-statistic and (adjusted) *p*-value for repeated tests reported. This methodology was also applied in comparing individuals presenting with/without OA at the time of OAI entry.

DDH allele and genotype frequency analyses: The Chi-Square statistic was used to compare observed (patient) to expected (1000 Genomes Project Population) rs4911178 allele and genotype frequencies, with 1 degree of freedom for two alleles, or 2 degrees of freedom for three genotypes. This was performed in two different ways. First, we compared data from DDH and 1000 Genomes Project individuals directly not accounting for differences in sample size (e.g., 113 DDH patient sets versus a 103 CHB sample set). Second, we compared data from DDH and 1000 Genomes Project individuals by recalculating allele and genotype counts using the observed allele frequency of the "A" variant and genotype frequency of the "A/A" genotype in the CHB set but scaled to 113 individuals (to match the sample number of DDH patients). DDH patient rs4911178 allele and genotype analyses" and presented in Supplementary Data 4.

Allele-specific expression: Pyrosequencing results for each SNP were used to calculate the allelic ratios of C57BL/6J (e.g., *GROW1* rs4911178 mouse orthologue "A" allele in the heterozygous state) to 129×1/SvJ (wild type allele). Each ratio of cDNA products found in heterozygous animals was then normalized by the ratio of WT C57BL/6J to 129×1/SVJ genomic products, amplified from known 1:1 mixtures of each sequence. The permutation test, a non-parametric measure, was used to determine significance between the wild type and heterozygous allelic expression ratio with the module in R.

Gene expression experiments: For each gene (*GDF5*, *CEP250*, and *UQCC1*) expression data were normalized relative to *GAPDH* house-keeping gene expression and compared between control and experimental condition (e.g., *GROW1* enhancer deletion or *GROW1* 13-bp rs4911178 variant position deletion). All data are presented as the mean ± SEM. Individual pairwise comparisons between control and experimental condition were analyzed by two-sample, two-tailed Student's *t*-test unless otherwise noted, with p < 0.05 regarded as significant. N numbers listed in figure legends (n = 3 biological replicates per comparison).

**Reporting summary**. Further information on research design is available in the Nature Research Reporting Summary linked to this article.

### Data availability

All the raw data used in this paper are included in Figs. (2–5), Supplementary figures (1–9), supplementary Data (1–6), as well as supplementary tables (1–15). All developmental human ATAC-seq sequencing data (raw sequencing FASTQ files and processed peak bed files) have been deposited on NCBI GEO under accession code GSE153260. All adult knee OA data from Liu et al.[57] can be found on NCBI Geo under accession code GSE108301. All OAI datasets can be found at https://oai.epi-ucsf.org.

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

## Acknowledgements

The authors would like to thank: Drs Michael Bruce, Daniel Brooks, and Mary Bouxsein for microCT work from the Imaging and Biomechanical Testing Core as well as Drs. Marie Demay, Gina Marie Grenga, and Yingzi Yang from the Histology & Histomorphometry Services Core both of the Center for Skeletal Research (NIH P30 AR066261) at the Massachusetts General Hospital; EpigenDx for allele-specific expression on mouse tissues; Applied StemCell for their effort in generating allelic replacement mice; Drs. Pardis Sabeti (Harvard University and Broad Institute), Li Zeng (Tufts University), and Mary Goldring (The Hospital for Special Surgery) for T/C-28a2 human chondrocyte and additional cell lines; the Harvard Genome Modification Facility for CRISPR-Cas9 targeting; the Harvard Bauer Core Facility for ATAC-seq assistance and sequencing; Dr. Steven Worthington (Harvard University) for statistics support and guidance; Dr. Guanyu Gong (Affiliated Zhongshan Hospital of Dalian University) for PCR assistance; and Drs. David Felson (Boston University), Steven Pregizer (Boston Children's Hospital), and members of the Capellini laboratory for critical insight and manuscript review. D.W.Z. was supported by the Thirteenth National Research and Invention Program of China (No. 2016YFC1102000) and the Dalian Science and Technology Innovation Fund Project (No. 2018J11CY030). A. M.K. was supported by grants from the NIH/NIAMS (R01AR065462) and the Faculty Council and Orthopaedic Foundation at Boston Children's Hospital. T.D.C. was supported by grants from NIH/NIAMS (1R01AR070139) for the mouse portion of this research and the Dean's Competitive Fund for the human ATAC-seq portion of this research at Harvard University. T.D.C. was supported by a grant from NSF (BCS1518596) for the mouse ATAC-seq portion of this research.

## Author contributions

T.D.C. conceived the project, designed the study, and supervised all aspects of the project. M.Y. and P.M. performed human ATAC-seq and downstream computational ATAC-seq analyses. T.D.C. performed GWAS/ATAC-seq computational intersections. D.R. performed joint trait/disease GWAS loci analyses, OAI patient sample processing, OAI genotype frequency analyses, and modularity analyses. T.D.C., P.M, and J.C. designed mouse genetic experiments, including *R4* and *GROW1* CRISPR-Cas9 gene editing, genotyping, and *LacZ* transgenesis. Z.L. performed *R4* mouse breeding experiments, and *R4* mutant micro CT preparations. P.M. performed *GROW1* mouse breeding experiments, mutant skeletal preparations, and micro CT preparations, and *GROW1* mouse allele-specific expression analyses, and micro CT analyses. A.M.K., G.P., and A.E. performed all morphometric analyses on *GROW1* and *R4* mutant mice micro CT data. G.P., A.E., and S.H. performed all morphometric analyses on OAI subjects under the guidance of A.M.K. A.M.K., and D.M. performed histologic assessment of knee OA based on OARSI scoring. A.M.K. performed all the relevant statistical analyses for morphometric data in mice and humans. P.M. performed human cell culture, CRISPR-Cas9 experiments, and qRT-PCR assays on the *GROW1* elements. P.M. and M.S. performed PITX1 over-expression experiments. P.M. performed with D.E.M., under the guidance of and in collaboration with V.R., micro CT imaging of allelic replacement mice. D.W.Z. diagnosed

and performed surgeries for DDH patients. D.W.Z. and Y.D. designed and supervised the medical imaging data analysis of DDH patients. L.L.C. performed surgeries of DDH, and worked with L.L. for demographic and morphometric data analysis of DDH patients. L.L.C. and L.L. collected surgical pathological specimens of DDH patients. Y.D. designed and supervised the genotyping of DDH surgical specimens. X.J.D., under the guidance of Y.D., integrated and analyzed genotyping and imaging data of DDH patients. T.D.C., A.M.K., and P.M. wrote the manuscript with input from all authors.

## Competing interests

The authors declare no competing interests.
