## [Peer Review File · Nature Communications]

Reviewers' Comments:

Reviewer #1:

Remarks to the Author:

The work from Muthuirulan et al dissects the complexities of genetic associations in humans. One of the baffling observations from these studies is the often pile up of multiple diseases in the same loci. Assuming that underlying these associations are regulatory variants that modify the properties of tissue-specific regulatory elements, one would not expect the pleiotropic effects of a single variant on multiple seemingly unrelated traits emerging from various tissues and distinct spatial specificities in the body. And yet, this extensive epistasis is what we commonly see. The investigators focus on the multiple associations of distinct Musculo-skeletal disorders in the vicinity of GDF5, a gene amply reported as mediating several of these GWAS disease phenotypes. Capitalizing on previous publications from their lab, they show that two variants in distinct enhancers, with distinct spatial specificities, result in altered GDF5 expression and in morphologic osteo-articular parameters. Their data posit that a possible resolution of the apparent paradox of extensive disease pleiotropy at the GDF5 locus is that distinct variants, each impacting distinct enhancers, may mediate susceptibility to multiple disorders.

The paper is interesting and there is considerable novelty in the development of the hypothesis of multiple variants per locus causing multiple phenotypes. There are several caveats, however. A major one is that lots of the data presented here are not novel, but rather have been shown in references 16 and 42, both the senior author. This redundancy significantly detracts from the novelty of the data presented here. Moreover, the oversimplified experimental strategy, assaying open chromatin in a single time point, identifying multiple candidate variants but focusing on only two that have already been previously extensively characterized, and the expansive interpretation of the implications of their findings temper enthusiasm with this paper. A revised version would benefit from being significantly shortened and focused on this exciting hypothesis, that needs to be also expanded to investigate if it holds true in other loci associated with Musculo-skeletal disorders or if this corresponds to an oddity of the GDF5 locus. My concerns and suggestions, in no order of importance are:

Some of the main findings here are not novel, having been reported by the senior author when they were in David Kingsley's lab (Capelini et al, Nat Gen 2017) or recently in their lab (Richard et al, Cell 2020). Figure 1B, which sets the stage for this paper is virtually identical to figure 5A of their Richard et al paper. That is published data and should not be reproduced in this paper as novel insights. Connected to the above, the molecular characterization of the impact of alternative alleles at rs4911178 (lines 197-218) describes how PITX1 has weaker affinity for the A allele. However, this is a well described finding in their earlier, Capellini et al 2017 paper: "Interestingly, the orthologous GROW1 sequence in humans contains a common derived SNP (rs4911178) at an otherwise highly conserved position. This change alters a predicted binding site for PITX1, a quintessential regulator of hindlimb growth and patterning in many animals⁴³ (Supplementary Fig. 15 and Supplementary Table 7). Experimental assays showed that the derived sequence change decreased PITX1 binding interactions in vitro (Supplementary Fig. 15 and Supplementary Table 7) and decreased GROW1 enhancer activity in both transgenic mice and human growth-plate chondrocytes (Fig. 4)." Fig1B shows the intersection of the 95% CI SNPs for OA and hip dysplasia and their intersection with ATAC-seq peaks While the two SNPs selected (rs4911178 and rs6060369) show overlapping between all three metrics above, there are numerous other credible SNPs that intersect with open chromatin for each of the diseases. What is the interpretation for this?

Most of these diseases are late-onset. Yet the open chromatin assay was done only in embryonic chondrocytes. While the justification offered (that at this stage they are homogeneous in cellular composition, it is possible that certain SNPs are altering the function of GDF5 enhancers later in life and have a significant impact on the susceptibility to OA and/or DDH. The authors need to acknowledge that their screen fails to account for the possibility that the allelic effects that lead to OA and DDH are ongoing, into adulthood.

There are numerous SNPs in the 95% CI for both diseases that map within ATAC-seq peaks. Are these enhancers? Do their allelic variants impact on GDF5 expression and may contribute to these diseases,

in addition to the at least 2 distinct variants they identify here?

The assessment of allele frequency differences between cases and controls (lines 153 to 160) seems intuitive given that this SNP is on the credible set of a haplotype associated with the disease. The frequency differences, shown in suppl. Table 5, can be stated in an abbreviated form in the text, making it more fluid.

In Fig.2A, why are the results contrasted with healthy individuals of Asian ancestry only? This needs to be contextualized in the text.

In the discussion, the authors state "Our current 377 findings suggest that GDF5 role in joint diseases is primarily developmental". This is a very intriguing idea, that late onset diseases might be due to embryonic, developmental gene dysregulation. But the design of the work in this manuscript does not support this conclusion. The authors only found developmental enhancers, because they only looked at chromatin accessibility data from embryonic chondrocytes. It is unclear if the phenotypic effects of the alleles in either enhancer would not be maintained through adulthood and compounded on any embryonic phenotype that they may impact. Thus, the authors need to temper their conclusion and raise this instead as a possibility that needs to be thought tested in future studies.

The authors make quite generalized conclusions about the genetic architecture of loci associated with complex traits. But it is unclear if their findings at GDF5 are instructive of other Musculo-skeletal disorders or, even less, to complex traits in general. Is the pleiotropic architecture of GDF5 in Musculo-skeletal disorders an oddity? The authors should expand on their search and look at all loci in these diseases, generating comprehensive data akin to the one presented in figure 1.

The tone of the discussion is at times unfocused and in other instances, hyperbolic. The repeated mention of disease alleles in "billions of people" gives an unnecessary impact to an otherwise trivial and mundane observation common to virtually every complex trait. Expanding the observations of this work as diagnostic screen tools or personalized medicine also seems far-fetched and directly at odds with the very observation by the authors that these variants are present in billions of people. I urge the authors to do a thorough rewriting of the discussion section, focusing on the novelty of this work, which is about genetic architecture of complex traits.

Reviewer #2:

Remarks to the Author:

I very much enjoyed reviewing this excellent paper that clearly advances our understanding of the functional relationship between non-coding genetic variation and joint shape. The analyses performed take advantage of cutting edge technologies to develop some truly novel insights into GDF5 function, a critical player in joint shape development, DDH and OA. I have some queries around the connections between the phenotype and genotype data, outlined below:

The analysis shown in fig 2a is not illuminating - a more vertical inclination is a diagnostic feature of human DDH versus a normally developed control hip. As such this figure does not advance the case for the genotype association or genetic functional mechanism, it simply says that the DDH patients have DDH and the controls did not by this anatomic criteria (a fact that we take 'as read'). The more appropriate Figure here would have been to show the association between carriage of the A allele and acetabular inclination angle across both cases and controls as a QTL (or similar). Without this analysis to confirm an association between rs4911178 genotype and this anatomic feature of human DDH, the downstream analyses (in humans) are based upon a less solid footing. The alternative is to simply delete 2a and its analysis and proceed with the interesting, but not clearly parallel mouse story, but this would weaken the clinical relevance. Figure 2b does show that the enhancer is in the right place anatomically (in mice) and at the right stage, but this is circumstantial unless the human association is demonstrated. 2c and d show that in mice there is a clinically relevant phenotype consistent with the story, but 2a needs to show the human equivalent in order for it all to hang together.

Figure 3a shows that GROW1 is a relevant region for regulation of Gdf5 expression and 3b that KO of rs4911178 is important within that region. 3c shows a clear differential effect of the "A" allele in decreasing gene expression in the mouse (as do 3f,g). However it does not show a convincing differential effect at the proximal femur more than the distal femur (where no phenotype is declared). The distal femur effect might visually be slightly less, but nonetheless is real. Can the authors present a proximal versus distal pairwise type analysis to show the allelic expression difference is site-specific? Again, in 3f the A/G pull-down effect appears similar between proximal and distal femur. Can a difference between the 2 be demonstrated?

Page 10 line 224: They have not actually demonstrated association of a more vertical acetabulum in the "A" allele-carrying DDH patients, only association with the "DDH" label. See my first point above.

Given that the functional effect of the allele also appears at the distal femur, it seems odd that there is no knee phenotype. This is clinically important, as most human femoral growth occurs at the distal growth plate and thus if functionally relevant we would expect to see a larger difference here. Perhaps there are other mechanistic features that modulate differential chondrocyte function between these sites?

R4 enhancer and functional effect of rs6060369: Figure 4b from the knee OAI data shows a neat, albeit small, allele dose effect on human knee shape, an effect that can be strongly amplified in engineered mice lacking the R4 enhancer. This kind of clinically-generated data would really help the arguments made in figure 3.

Why were the OA mechanistic analyses limited to knee? GDF5 is also associated with hip OA in GWAS, and it would be interesting to understand the relation of the enhancer regions to hip OA variants (although noting that there is shared heritability between DDH and hip OA).

Figure 5 neatly draws together the site specific effects of these non-coding regulatory regions in GDF5, and suggest that allelic homozygosity at each site appears to account for almost 2/3 of the effect of region KO, and that GDF5 expression variation seems to account for about 50% of the observed phenotypic variation. Could the authors demonstrate this quantitatively by regression analysis?

The access to embryonic anatomic human tissue for ATAC-seq is a major focus of this work and a key novel functional tool for the group. Although day 67 is a very useful time to study chromatin accessibility, this timepoint is also opportunistic for the availability of tissue. As we would expect chromatin accessibility to vary across gestation, presumably with organ development, some further comment in the discussion on how this opportunity coincided with the the relevant phase of limb development would help to contextualise the ATAC-seq data. Was there access to embryonic tissue at other stages of development, and how does this differ to confirm that day 67 is optimal?

If the message is about human development, doesn't the mouse ATAC-seq data somewhat limit the scope of interpretation to those regulatory regions that are conserved? Other ATAC-seq regions appear to intersect with the SNP data in humans only in fig 1., particularly that near GROW1, but outside of these annotated ATAC-seq regions. It would be useful to understand which SNPs are intersecting here. Perhaps an item for discussion in the study limitations.

Mark Wilkinson

Point-by-Point Response to Reviewers

We thank both Reviewers for their detailed examination of our work and providing excellent constructive comments. We have addressed their comments below and made many emendations to Main Manuscript, Supplementary Information Document, Tables, and Figures. We believe our manuscript is much improved as a result. Below, we note all Reviewer comments are identified in quotes, while our responses are in blue font.

Reviewer #1:

“The work from Muthuirulan et al dissects the complexities of genetic associations in humans. One of the bafflingly observations from these studies is the often pile up of multiple diseases in the same loci. Assuming that underlying these associations are regulatory variants that modify the properties of tissue-specific regulatory elements, one would not expect the pleiotropic effects of a single variant on multiple seemingly unrelated traits emerging from various tissues and distinct spatial specificities in the body. And yet, this extensive epistasis is what we commonly see. The investigators focus on the multiple associations of distinct Musculo-skeletal disorders in the vicinity of GDF5, a gene amply reported as mediating several of these GWAS disease phenotypes. Capitalizing on previous publications from their lab, they show that two variants in distinct enhancers, with distinct spatial specificities, result in altered GDF5 expression and in morphologic osteo-articular parameters. Their data posit that a possible resolution of the apparent paradox of extensive disease pleiotropy at the GDF5 locus is that distinct variants, each impacting distinct enhancers, may mediate susceptibility to multiple disorders.”

“The paper is interesting and there is considerable novelty in the development of the hypothesis of multiple variants per locus causing multiple phenotypes.”

We thank this Reviewer for an excellent summary and noting our work has considerably novelty.

“There are several caveats, however. A major one is that lots of the data presented here are not novel, but rather have been shown in references 16 and 42, both the senior author. This redundancy significantly detracts from the novelty of the data presented here.”

Here, we briefly note that we have reorganized the paper and each of its main sections by substantially reducing content and callouts to previous findings from Capellini et al., *Nat Gen* 2017 and Richard et al., *Cell* 2020. We believe these changes now allow the reader to more clearly identify the novel functional and computational aspects of the work. These changes are also specifically addressed in the sections below where Reviewer #1 provides more detail.

“Moreover, the oversimplified experimental strategy, assaying open chromatin in a single time point, identifying multiple candidate variants but focusing on only two that have already been previously extensively characterized, and the expansive interpretation of the implications of their findings temper enthusiasm with this paper.”

We have addressed this comment in a number of ways, and note that the details discussed in the following pages are now presented in an expanded Supplementary Information section entitled “ATAC-seq peak and risk variant intersection to whittle-down the *GDF5-UQCCI* DDH and knee OA association intervals”, and elsewhere when indicated. First, we note that our single human embryonic time-point ATAC-seq data overlaps with ATAC-seq data at different human gestational timepoints (also please see response to Reviewer 2 comment). Second, we now use an additional ATAC-seq dataset from osteoarthritis patient knee cartilage (Liu et al., 2018) to intersect with knee OA variants. Third, we provide details from GTEx eQTL data on each variant discussed in the paper. Fourth, we provide more detailed descriptions on our prior work using a functional reporter tiling array screen across the *GDF5* locus, an approach that also involved the use of bacterial artificial chromosomal (BAC) rescue experiments *in vivo* to narrow down putative variants to the downstream *GDF5* regulatory sequences. All of these additions, and others noted below help strengthen and justify our variant choice.

Additionally, we have improved the Manuscript and Supplementary Information text by discussing that developmental dysplasia of the hip (DDH) is a skeletal disorder that arises during prenatal development, and so our ATAC-seq approach on human embryonic tissues is appropriate. But as noted above, we have also modified the text describing the use of an additional knee OA patient cartilage ATAC-seq dataset to find putative causal variants.

We would like to emphasize three additional points regarding the novelty of this work. First, existing, large scale ENCODE and Roadmap Epigenomics Project datasets shed very little light on causality at the locus and at other skeletal GWAS loci because these projects have not focused on characterizing the epigenome of *in vivo* collected skeletal cell types either by pooling cells or at site-specific anatomical locations. While we use ENCODE and Roadmap Epigenomics data to assist in our experimental work, they do not inform on musculoskeletal disease/trait biology as they currently stand. We therefore have taken the reigns to improve epigenome profiling of rare human skeletal samples, and we have presented a rare ATAC-seq dataset here as a novel contribution. Second, the work performed for this paper is already quite novel, extensive, and has led to important insights about complex disease/trait genetics. For example, there are currently well over 200,000 GWAS loci (spanning millions of variants) that have been detected across studies and the number of validated functional non-coding variants is less than 20 (Buniello et al., 2019). In this work, we identified and functionally validated for the first time one variant that causes DDH, but our approach also allowed us to strengthen the importance of another variant as causal specifically for knee OA only. Thus, by doing so we demonstrate that two separate causal

variants reside on the same risk haplotype. To this end, our extensive phenotyping of lines previously published (from Richard et al., *Cell* 2020; Capellini et al., *Nat Gen* 2017) is essential to the demonstration of allelic causality for both variants. Yet, we also developed two new lines for this manuscript (see below), and via phenotyping and other detailed molecular analyses reveal the causal variant for DDH. Third, we have augmented our analyses following the Reviewer's wonderful suggestion to explore whether the pattern we observed at *GDF5* holds for other musculoskeletal disease loci, and by doing so have also contributed additional novelty to the work (see below).

“A revised version would benefit from being significantly shortened and focused on this exciting hypothesis, that needs to be also expanded to investigate if it holds true in other loci associated with Musculo-skeletal disorders or if this corresponds to an oddity of the *GDF5* locus”

We have significantly improved the manuscript text and have made it more streamlined in all three sections (Introduction, Results, Discussion). These changes reflect the suggestion by the Reviewer to explore whether what holds for *GDF5* is also found at other loci (see below).

My concerns and suggestions, in no order of importance are:

“Some of the main findings here are not novel, having been reported by the senior author when they were in David Kingsley's lab (Capellini et al, *Nat Gen* 2017) or recently in their lab (Richard et al, *Cell* 2020). Figure 1B, which sets the stage for this paper is virtually identical to figure 5A of their Richard et al paper. That is published data and should not be reproduced in this paper as novel insights.”

We thank Reviewer #1 for making these points, and by-doing-so urging us to improve the manuscript and thus highlight better its novelty. Aside from the additional datasets and new analyses on modularity, we thought it would be necessary to better clarify below the contributions from both cited works (Capellini et al., *Nat Gen* 2017; Richard et al., *Cell* 2020) and in the context of each discuss the novel contributions in our resubmitted (Muthuirulan et al) manuscript

In Capellini et al., *Nat Gen* 2017, we characterized the entire ~1 kb GROW1 enhancer and its expression in chondrocyte growth plates, and by knocking out the entire enhancer in mice show its effects in limb length and femoral neck length. In that paper's context we report on a genetic variant associated with height (rs4911178), under natural selection, and we test the effects of the variant using *in vitro* luciferase reporter assays (in human CHON-02 cells) and via lacZ transgenesis in the mouse. We therefore never modelled the single base-pair variant in this regulatory enhancer directly in mice nor for its effect on DDH. In this Muthuirulan et al., we have engineered two new mouse lines and used them to address DDH versus knee OA causality.

The first is a stable lacZ line in which the lacZ reporter gene is under the control of the human GROW1 regulatory sequence. This line is used to show *in vivo* how the human enhancer drives expression in hip (and not knee) and therefore relates to DDH. The other line is a humanized single base-pair replacement mouse line for the regulatory “A” risk variant in GROW1. This type of engineering within a regulatory enhancer is not straightforward nor done often for single allelic changes. We then use this line to show direct effects of the variant change on *Gdf5* expression *in vivo* in the hip. Using this mouse line, we also show that there is PITX1 binding at the variant position and that the “A” reduces this binding *in vivo* (see response to comment below). Using this line, we then show the risk allele causes a marked phenotypic impact on hip morphology and causes DDH phenotypes. We then show that this line, as well as the previously reported GROW1 enhancer knockout line, do not impact the knee. In the context of these DDH centered studies we also performed direct patient analyses, both at the genotypic and phenotypic level, that show the same directional and disease related phenotypes between patients and humanized mice.

In Richard et al., *Cell* 2020, we characterized the entire R4 knee enhancer element, its expression in the articular chondrocyte regions, and by knocking it out in mice show its effects on knee joint morphology. In that paper’s context we also report on a genetic variant associated with knee OA (rs6060369) and we test the variant in vitro using reporter assays, as well as in vivo by engineering humanized single base-pair replacement mice to examine its effects on gene expression and phenotype. In Muthuirulan et al., we have now used these prior engineered R4 enhancer null and “T” single base-pair replacement mice to show the lack of hip phenotypes, which has never been reported, as well as to identify phenotypes at 1 year of age in the knee which are directly related to knee OA, which was never reported. We additionally provide direct patient analyses, both at the genotypic and phenotypic level, that show the same directional and disease related phenotypes between patients and humanized mice.

Regarding Muthuirulan et al. Figure 1B, while it appears to resemble that shown in Richard et al., *Cell* 2020 (Figure 5A), the datasets used are nearly all different save only two repeated experimental tracks (one track is a published knee OA GWAS variant set, and another is a track showing the locations of previously reported *GDF5* regulatory elements). All other experimental tracks shown in Muthuirulan et al. Figure 1B are new, including a more comprehensive ATAC-seq dataset from a different human gestational timepoint. We note in the Supplementary Information document that these ATAC-seq peaks overlap with ATAC-seq data at a different gestational time point (from Richard et al., 2020). We believe this figure is important to keep in the main text as it illustrates the intersections between experimental dataset tracks and GWAS variants. We have now updated an Extended Data Fig. 2 to include an additional track showing ATAC-seq regions from aging patient knee OA tissue. This was a published ATAC-seq data (Liu et al., 2018) from a group in Japan examining knee OA patient cartilage. Another reason we

feel it is necessary to include Figure 1B, is that as we have now expanded on the genome-wide analysis of modularity (as recommended by this Reviewer), we feel this locus-specific view at *GDF5* helps to show the reader the modularity patterns we are observing at other loci.

“Connected to the above, the molecular characterization of the impact of alternative alleles at rs4911178 (lines 197-218) describes how PITX1 has weaker affinity for the A allele. However, this is a well described finding in their earlier, Capellini et al 2017 paper: “Interestingly, the orthologous *GROW1* sequence in humans contains a common derived SNP (rs4911178) at an otherwise highly conserved position. This change alters a predicted binding site for PITX1, a quintessential regulator of hindlimb growth and patterning in many animals⁴³ (Supplementary Fig. 15 and Supplementary Table 7). Experimental assays showed that the derived sequence change decreased PITX1 binding interactions *in vitro* (Supplementary Fig. 15 and Supplementary Table 7) and decreased *GROW1* enhancer activity in both transgenic mice and human growth-plate chondrocytes (Fig. 4).”

We thank the Reviewer for discussing this point on PITX1 binding. While it is true that in Capellini et al., *Nat Gen* 2017 we first computationally predict PITX1 experimental binding and then cite experimental evidence to this end, we note that the cited experimental evidence is entirely *in vitro* and *in silico* derived. It should not be considered true demonstration of binding at the locus by PITX1. Specifically, the data in Capellini et al., 2017 comes from Newberger and Bulyk et al., 2009 (UniProbe database), which consists of an *in vitro* high-throughput binding affinity screen where PITX1 and hundreds of other transcription factors have been applied to a chip containing all 8- or 10-mers base-pair combinations, with 8-mers having any level of binding affinity recorded. In other words, the experimental binding we had cited is essentially indicating whether PITX1 was bound to the 8-mer in a completely sterile *in vitro/in silico* context.

In this Muthuirulan et al. manuscript we performed chromatin immunoprecipitation (ChIP) using an antibody to PITX1 on *in vivo* collected chondrocytes from the relevant anatomical location (proximal femur), and from humanized mice heterozygous for the risk allele (i.e. “A/G”). We then discovered actual binding *in vivo*. Moreover, because these ChIP experiments permit the sequencing of pulled-down amplicons, we could then demonstrate experimentally in the correct tissue/cell type and in the context of the correct controls, that the sequence with the “A” allele more weakly binds PITX1 than the sequence with the “G” allele. We then performed PITX1 over-expression studies on chondrocytes extracted from the relevant anatomical region from these heterozygous “A/G” mice and show it augments the effect of binding on the “G” but not “A” allele. This is a gold standard approach to show what is the mechanism of action at the locus. Overall, we feel that these experiments and others should be highlighted as they demonstrate mechanism of action. We note that only a handful of true causal non-coding variants identified for any disease (from hundreds of thousands of GWAS loci) have been shown to have

this level of demonstration of *in vivo* biochemical binding by a transcription factor and the effects of the variant *in vivo*, let alone *in vitro*.

“Fig1B shows the intersection of the 95% CI SNPs for OA and hip dysplasia and their intersection with ATAC-seq peaks. While the two SNPs selected (rs4911178 and rs6060369) show overlapping between all three metrics above, there are numerous other credible SNPs that intersect with open chromatin for each of the diseases. What is the interpretation for this?”

In a newly updated Supplementary Information section entitled “ATAC-seq peak and risk variant intersection to whittle-down the GDF5-UQCC1 DDH and knee OA association intervals” we provide detailed discussion of each variant intersection. As mentioned, we have also expanded it to include data from an additional ATAC-seq knee OA dataset, GTEx eQTL information, and information on previous functional studies on the locus. In all contexts, we provide insight as to why we examined rs4911178 and rs6060369. We also want to indicate that since DDH and knee are diseases involving three-dimensional structure/function relationships that arise over time, that variants residing in functional orthologous regulatory sequences in the mouse most appropriately facilitate the assessment of their functional impacts. Variants not residing in these sequences are more difficult to model, and given their expense, are also extremely risky. However, in the new manuscript we also do temper our discussion on our findings on rs6060369 “T” allele and knee OA stating that we cannot rule out the effects of other variants as contributing in part to knee OA disease risk.

“Most of these diseases are late-onset. Yet the open chromatin assay was done only in embryonic chondrocytes. While the justification offered (that at this stage they are homogeneous in cellular composition, it is possible that certain SNPs are altering the function of GDF5 enhancers later in life and have a significant impact on the susceptibility to OA and/or DDH. The authors need to acknowledge that their screen fails to account for the possibility that the allelic effects that lead to OA and DDH are ongoing, into adulthood.”

We thank Reviewer #1 for making this point and have improved the text to make our points clearer, and to specifically address this issue as it relates to knee OA. It is known that DDH is of developmental (embryonic and/or fetal) origin and that it results from altered chondrocyte and joint biology. We also know that developmental effects caused by disruptions to *GDF5* (coding sequences) cause *in utero* and early postnatal hip defects (cited in Muthuirulan et al.). Therefore, our ATAC-seq datasets generated on chondrocytes from the developing hips of human embryos during the stage when hip joints are forming (and in stage-matched mouse embryos) are highly relevant to this phenotype to identify causal variants.

Regarding knee OA, we have strong evidence, as presented in Richard et al., *Cell* 2020, that knee OA risk at *GDF5* is governed by shape alterations, and that these arise developmentally. We

know that developmentally *GDF5* knockout humans and mice have knee shape defects, and we know from novel data presented in Muthuirulan et al., that human knee OA patients have knee shape defects that correlate with the “T” allele at rs6060369 in the *R4* enhancer. We therefore targeted our work on the developmental effects at *GDF5* as being key driver of OA at the locus. However, since the *R4* enhancer is active post-natally (Richard et al., *Cell* 2020) and that it overlies ATAC-seq data from OA human knee cartilage (as we have now provided intersections in Supplementary Information), the “T” variant could continue to disrupt enhancer function throughout life and contribute to knee OA disease. We have modified the text to make sure this point is noted by stating in the Results “We previously showed that the human risk “T” variant when modeled directly in mice (i.e. in R4rs6060369-T/rs6060369-T allelic replacement mice) impacts *Gdf5* expression and knee shape early in life with a potential role later. However, whether the allele causes more drastic shape defects and OA later in life remains unstudied, and it is unknown if it impacts hip biology.”

“There are numerous SNPs in the 95% CI for both diseases that map within ATAC-seq peaks. Are these enhancers? Do their allelic variants impact on *GDF5* expression and may contribute to these diseases, in addition to the at least 2 distinct variants they identify here?”

We politely direct this Reviewer to our answers above, and to our newly updated Supplementary Information section entitled “ATAC-seq peak and risk variant intersection to whittle-down the *GDF5-UQCCI* DDH and knee OA association intervals.”

“The assessment of allele frequency differences between cases and controls (lines 153 to 160) seems intuitive given that this SNP is on the credible set of a haplotype associated with the disease. The frequency differences, shown in suppl. Table 5, can be stated in an abbreviated form in the text, making it more fluid.”

We have now made the requested change, and the new text reads “In patients, the “A” allele frequency was 84.5%, a highly significant enrichment (an 8.3% increase) compared to population controls at 76.2% ($p < 0.005$).”

“In Fig.2A, why are the results contrasted with healthy individuals of Asian ancestry only? This needs to be contextualized in the text.”

We have improved this entire analysis according to Reviewer #2 comments, but have retained Figure 2A as a supplemental figure. The new analyses we performed shows that DDH patients with the “A/A” (DDH homozygous risk) genotype have significantly different and more exacerbated acetabular inclination angles (and other measures) than “A/G” heterozygous controls, indicating that the “A” allele correlates with relevant worsening shape parameters in the patient cohort. This matches the direction of effect as seen in mice having “A/A” versus “A/G”

and “G/G” genotypes. We now have included the original Figure 2A as a supplemental figure panel (Extended Data Fig. 3) to demonstrate that compared to healthy controls from a variety of east Asian populations, individuals with DDH have higher acetabular inclination angles (with those with “A/A” genotypes even higher worsening angles).

“In the discussion, the authors state “Our current 377 findings suggest that GDF5 role in joint diseases is primarily developmental”. This is a very intriguing idea, that late onset diseases might be due to embryonic, developmental gene dysregulation. But the design of the work in this manuscript does not support this conclusion. The authors only found developmental enhancers, because they only looked at chromatin accessibility data from embryonic chondrocytes. It is unclear if the phenotypic effects of the alleles in either enhancer would not be maintained through adulthood and compounded on any embryonic phenotype that they may impact. Thus, the authors need to temper their conclusion and raise this instead as a possibility that needs to be though tested in future studies.”

We thank Reviewer #1 for this comment and ask them to see our responses above. We feel we have also tempered our conclusions in a number of places including in a new expanded Supplemental Information section entitled “ATAC-seq peak and risk variant intersection to whittle-down the *GDF5-UQCCI* DDH and knee OA association intervals.”

“The authors make quite generalized conclusions about the genetic architecture of loci associated with complex traits. But it is unclear if their findings at GDF5 are instructive of other Musculo-skeletal disorders or, even less, to complex traits in general. Is the pleiotropic architecture of GDF5 in Musculo-skeletal disorders an oddity? The authors should expand on their search and look at all loci in these diseases, generating comprehensive data akin to the one presented in figure 1”.

We thank the Reviewer for making this excellent comment and have now addressed this concern computationally by first defining better what modularity is in the context of our skeletal site-specific ATAC-seq datasets and applying this metric across the genome. We defined non-modular loci are those where at each regulatory element in the locus the presence of an accessibility signal does not differ from all other elements in the locus (for example, if we detected 13 regulatory regions in a locus, all 13 would show the same exact tissue accessibility signals). We then defined modular loci are those where there is variation in accessibility from one region to the next region in a locus (for example, if we detected 7 regulatory regions in a locus, several may share accessibility in 2 of 5 tissues, whereas several others share accessibility in 3 of 5 or 1 of 5, etc.). Using these definitions, we then assess loci with singular or multiple musculoskeletal GWAS disorder/trait signals to see if they behave similarly to *GDF5*. We found that over three-quarters of loci with multiple GWAS associations have modularity patterns like *GDF5*, albeit these modularity patterns are also found across the genome. We have updated the

Introduction, Results, and the Discussion to address these new findings and their implications to understanding the genetic architecture of complex traits. We have also added a relevant Methods section.

The tone of the discussion is at times unfocused and in other instances, hyperbolic. The repeated mention of disease alleles in “billions of people” gives an unnecessary impact to an otherwise trivial and mundane observation common to virtually every complex trait. Expanding the observations of this work as diagnostic screen tools or personalized medicine also seems far-fetched and directly at odds with the very observation by the authors that these variants are present in billions of people. I urge the authors to do a thorough rewriting of the discussion section, focusing on the novelty of this work, which is about genetic architecture of complex traits.

We thank this Reviewer for these comments and have reworked the Discussion to highlight how our work impacts understanding the genetic architecture of complex traits. Given our expertise, we have taken an evolutionary angle to the new Discussion section. We have also substantially shortened the content relating to *GDF5*, its prior work, and its medical ramifications. However, we also feel that given the relevance of *GDF5* (and many loci) to the biomedical field, that it is important to retain some language for the broader readership. Please see the newly revised Discussion.

Reviewer #2 (Remarks to the Author):

“I very much enjoyed reviewing this excellent paper that clearly advances our understanding of the functional relationship between non-coding genetic variation and joint shape. The analyses performed take advantage of cutting edge technologies to develop some truly novel insights into GDF5 function, a critical player in joint shape development, DDH and OA. I have some queries around the connections between the phenotype and genotype data, outlined below:”

We thank the Reviewer for these terrific comments and appreciate his suggestions, all of which we have addressed.

“The analysis shown in fig 2a is not illuminating - a more vertical inclination is a diagnostic feature of human DDH versus a normally developed control hip. As such this figure does not advance the case for the genotype association or genetic functional mechanism, it simply says that the DDH patients have DDH and the controls did not by this anatomic criteria (a fact that we take 'as read').

We have removed Figure 2a from the main text, and instead placed it in the Supplementary Information document for reasons discussed below. We have also performed new analyses which are discussed in response to the next comment.

“The more appropriate Figure here would have been to show the association between carriage of the A allele and acetabular inclination angle across both cases and controls as a QTL (or similar). Without this analysis to confirm an association between rs4911178 genotype and this anatomic feature of human DDH, the downstream analyses (in humans) are based upon a less solid footing. The alternative is to simply delete 2a and its analysis and proceed with the interesting, but not clearly parallel mouse story, but this would weaken the clinical relevance. Figure 2b does show that the enhancer is in the right place anatomically (in mice) and at the right stage, but this is circumstantial unless the human association is demonstrated. 2c and d show that in mice there is a clinically relevant phenotype consistent with the story, but 2a needs to show the human equivalent in order for it all to hang together.”

Unfortunately, we do not have genome-wide SNP genotyping or complete genome sequencing from our DDH patient cohort and so we cannot perform a GWAS; plus, our sample size is extremely low. However, we have addressed this Reviewer’s comments by asking whether DDH measures associate with the “A” allele in our patient cohort. We asked whether we see differences in acetabular inclination angle, acetabular index, and center-edge angle between patients who are “A/A” (homozygous risk) versus “A/G” (heterozygous risk). We indeed saw significant changes in these measures. For example, patients with the “A/A” genotype show an increased acetabular inclination angle in the direction of increased and worsening DDH, a

direction we also observed in mice with “A/A” genotypes compared to “A/G” and “G/G” mice. These data are now presented as a new Fig. 2a. We retained the original Fig 2a as a new panel in Extended Data Fig. 3. We have updated legends, and Methods accordingly.

Figure 3a shows that GROW1 is a relevant region for regulation of *Gdf5* expression and 3b that KO of rs4911178 is important within that region. 3c shows a clear differential effect of the “A” allele in decreasing gene expression in the mouse (as do 3f,g). However, it does not show a convincing differential effect at the proximal femur more than the distal femur (where no phenotype is declared). The distal femur effect might visually be slightly less, but nonetheless is real. Can the authors present a proximal versus distal pairwise type analysis to show the allelic expression difference is site-specific? Again, in 3f the A/G pull-down effect appears similar between proximal and distal femur. Can a difference between the 2 be demonstrated?

We agree with this Reviewer’s points. We do not want to downplay the importance of the decreased *Gdf5* expression and PITX1 binding in the distal femur due to the “A” allele (rs4911178), but more so indicate that in the relevant proximal femur tissue those changes are important for DDH. However, as the Reviewer rightfully alluded to in a comment below, we actually do see a distal femur phenotype in single base-pair replacement mice and that is a decrease in femur length. We found that femur length was reduced in “A/A” mice compared to “A/G” and “G/G” mice (see figure below). We did not see changes in tibial length (where effects on *Gdf5* expression and PITX1 binding were markedly weaker), so it is a localized effect. Given the importance of the distal femoral growth plate in driving length of the femur, we note this is evidence of an effect on the distal femur of the “A” allele which does decrease *Gdf5* expression and does decrease PITX1 binding in that tissue. Coincidentally we also saw no length changes in our knee joint enhancer R4 “T” single base pair enhancer replacement mice (not shown), which indicates that we can uncouple this height signal and identify the appropriate tissue (i.e. growth plate versus epiphyseal cartilage). These femur length data were not included in this manuscript as we had intended to present them in a paper in coordination with a current collaboration with the GIANT Consortium, briefly revealing that with the newest unpublished height GWAS, the *GDF5* association can be functionally pinpointed using these methods. We would be glad to add these findings to the manuscript if requested. We will finally note that below the distal femur growth plate (i.e., approaching and within the knee joint proper) we saw no effect of the “A” allele on metric measurements taken (as reported in the original manuscript).

Page 10 line 224: They have not actually demonstrated association of a more vertical acetabulum in the "A" allele-carrying DDH patients, only association with the "DDH" label. See my first point above.

Please see our comments above.

Given that the functional effect of the allele also appears at the distal femur, it seems odd that there is no knee phenotype. This is clinically important, as most human femoral growth occurs at the distal growth plate and thus if functionally relevant we would expect to see a larger difference here. Perhaps there are other mechanistic features that modulate differential chondrocyte function between these sites?

Please see our comments above.

R4 enhancer and functional effect of rs6060369: Figure 4b from the knee OAI data shows a neat, albeit small, allele dose effect on human knee shape, an effect that can be strongly amplified in engineered mice lacking the R4 enhancer. This kind of clinically-generated data would really help the arguments made in figure 3.

We thank the Reviewer for this comment. Please see our response to one of the earlier comments above, where we discuss the new analyses we have performed on the patient DDH dataset.

Why were the OA mechanistic analyses limited to knee? GDF5 is also associated with hip OA in GWAS, and it would be interesting to understand the relation of the enhancer regions to hip OA variants (although noting that there is shared heritability between DDH and hip OA).

When we examined the R4 knee enhancer knockout mice as well as R4 single base pair rs6060369 “T/T” mice, we observed no changes in the hip at either P30 or P365. We also did not observe signs of hip OA in rs6060369 “T/T” mice. We therefore believe that the knee phenotype in these mice is mild enough to not cause hip OA, albeit whether the variant causes hip OA much later in life (like at 2 years in mice) is unknown. But, hip OA is not a guaranteed genetic consequence of knee OA.

We had also wanted to assess whether GROW1 knockout as well as “A/A” replacement mice get hip OA (given their DDH phenotypes). In March 2020, the corresponding author’s (Capellini) laboratory was closed due to COVID-19. Upon closure, a number of issues arose, most notably the loss of several long term experiments. We had two underway for this project to determine whether mice harboring human genetic variants (at these regions) develop osteoarthritis at 1 and 2 years of age. Each experiment involved raising over 20 separate cages of mice (each with three different genotypes per enhancer variant) to 1 and 2 year time-points to study their knee and hip joints for signs of osteoarthritis. Given the complete uncertainty regarding mouse facility access, we had to cull all of these mice directly before closure in order to alleviate the burden on an already stressed animal facility staff (we were also not permitted to enter the campus and facilities). Besides the financial impact of this much earlier than expected culling, the scientific impact was the loss of an adequately powered and well-designed multi-year experiment. We will save these experiments for more grant funding as it will now take years to generate new data, and to help determine whether hip OA GWAS associations at *GDF5* are linked to this “A” allele at rs4911178.

“Figure 5 neatly draws together the site specific effects of these non-coding regulatory regions in *GDF5*, and suggest that allelic homozygosity at each site appears to account for almost 2/3 of the effect of region KO, and that *GDF5* expression variation seems to account for about 50% of the observed phenotypic variation. Could the authors demonstrate this quantitatively by regression analysis?”

We have modified Figure 5 and added an additional figure and details to show that homozygosity on average results in 1.7 fold (70%) more change in anatomy. We have also added the regression coefficient (beta) to show that for every 1% reduction in *Gdf5* expression there is 0.57% change in anatomy. These details are also added as Figure 5 a + b in the main body of the manuscript.

“The access to embryonic anatomic human tissue for ATAC-seq is a major focus of this work and a key novel functional tool for the group. Although day 67 is a very useful time to study chromatin accessibility, this time point is also opportunistic for the availability of tissue. As we would expect chromatin accessibility to vary across gestation, presumably with organ development, some further comment in the discussion on how this opportunity coincided with the relevant phase of limb development would help to contextualize the ATAC-seq data. Was there access to embryonic tissue at other stages of development, and how does this differ to confirm that day 67 is optimal?”

We have addressed this issue by reworking some of the text in our detailed descriptions and analyses in the Supplementary Information document. In a nutshell we found that the accessibility patterns across the locus hold for earlier gestational timepoint data (E59, published) as well as at E54 (unpublished). We have not formally added the E54 data since it is used in a manuscript currently under review addressing the evolution of the human pelvis (from chimpanzees) on a genome-wide level. That manuscript has no overlap with this work. Unfortunately, we cannot get access to earlier or later stage embryonic/fetal tissues for practical and ethical reasons.

“If the message is about human development, doesn't the mouse ATAC-seq data somewhat limit the scope of interpretation to those regulatory regions that are conserved? Other ATAC-seq regions appear to intersect with the SNP data in humans only in fig 1., particularly that near *GROW1*, but outside of these annotated ATAC-seq regions. It would be useful to understand which SNPs are intersecting here. Perhaps an item for discussion in the study limitations.”

We thank the Reviewer for this observation and remark. We note that our intersections of European and Asian DDH GWAS variants and human embryonic ATAC-seq data only found four variant intersections (each of which is discussed in the Supplementary Information document and presented in Table S4). Regarding the region around *GROW1*, in Capellini et al.,

2017 *Nature Genetics*, we had performed an enhancer tiling screen in a ~21 KB window around GROW1. In that screen we tested five adjacent/contiguous constructs (including GROW1) using lacZ transgenesis in the mouse, but testing the human as well as the mouse sequence. We saw no reproducible expression pattern in joints or long bones (or even connective tissues) by any human or mouse sequence save the human/mouse GROW1 sequence, thereby indicating that there are likely no active developmental regulatory enhancers in that window. It also meant that additional variants flanking rs4911178 and the GROW1 enhancer in this 21 KB interval are not likely enhancer variants in tissues relevant to height (in that study) and DDH (this study). We believe this is reflected in our open chromatin data from humans and mice around GROW1. However, as shown in the figure below, there is an ATAC-seq peak in human tissues located downstream of GROW1 (a black box to the right of the first long red vertical line on the left noting rs4911178), but we note there is no DDH variant in this peak. That sequence could also be some form of repressor or have other unknown functionality since it overlapped with our tiling screen but did not show direct enhancer activity. Nevertheless, in the new Supplementary Information document we made sure to discuss all additional variant intersections in the locus.

Overall, we hope that our comments and additional data and textual edits to the manuscript have satisfied this Reviewer - Thanks Mark Wilkinson!

Reviewers' Comments:

Reviewer #1:

Remarks to the Author:

The authors addressed my comments satisfactorily, the paper is clearer, more focused and improved.

Reviewer #2:

Remarks to the Author:

My queries have been fully addressed in the revised submission materials.

Mark Wilkinson

Reviewer #3:

Remarks to the Author:

This manuscript was a pleasure to read and the authors have done an excellent job on the revision in response to the two original reviewers. I was asked to provide an additional opinion relating to mouse models of joint disease. I have only a couple of comments that can be addressed in a minor revision.

1. The targeted genetics approach to model each variant in humanized mice was used to support the findings on joint specificity of GDF5 associations with human joint abnormalities, human DDH and knee osteoarthritis, respectively. The analyses in the mice focusing on changes in joint morphology provide further evidence that "knee OA risk at GDF5 is governed by shape alternations" and it is clear that the human data are from patients with radiologically/MRI diagnosed OA. However, it does not appear that the mice were histologically assessed to determine whether the variant affected cartilage damage as a hallmark of OA. The authors are correct in assuming that there are effects of the variants "on key joint-specific morphological features leading to altered biomechanics and stability" (page 16), which would lead to OA, and the influence of subchondral bone changes on cartilage is well accepted. The bone histomorphometry measurements were performed at P30 in the mice with the DDH-related variant and at 1 year of age in the mice with the knee OA-related variant, so the latter might show osteophytes, as another hallmark of OA. In any case, some note about how structural changes due to either variant in bone, which can adapt, would lead inevitably to OA changes in cartilage, which cannot adapt, would be warranted.

2. Page 9, Fig. 3a, b, and d: It should be mentioned in the text (although indicated in the legend) that the human chondrocytes used for the ChIP assays were a chondrocyte cell line (T/C-28a2). Upon first reading, since this followed immediately after reference to DDH patients, it appeared that primary chondrocytes from patients might have been used. Although it is well known that primary cells do not generally provide sufficient numbers of cells for ChIP assays, it might be mentioned in the Methods as a justification for using the cell line.

Point-by-Point Response to Reviewers (2nd Revision)

We thank all Reviewers for their detailed examination of our work and providing excellent constructive comments. Reviewer #1 and #2 had no additional inquiries. Below, we have addressed Reviewer #3 comments and made emendations to Main Manuscript, Supplementary Information Document, Tables, and Figures. We believe our manuscript has improved as a result. Below, we note all Reviewer comments are identified in quotes, while our responses are in blue font.

Reviewer #1:

“The authors addressed my comments satisfactorily, the paper is clearer, more focused and improved.”

We thank Reviewer #1 and their critical assessment of our work. Your insights have significantly strengthened the manuscript.

Reviewer #2:

“My queries have been fully addressed in the revised submission materials.”

We thank Reviewer #2 and their critical assessment of our work. Your insights have significantly strengthened the manuscript.

Reviewer #3:

“This manuscript was a pleasure to read and the authors have done an excellent job on the revision in response to the two original reviewers. I was asked to provide an additional opinion relating to mouse models of joint disease. I have only a couple of comments that can be addressed in a minor revision.”

We thank Reviewer #3 for the kind words and providing excellent comments as discussed below.

“1. The targeted genetics approach to model each variant in humanized mice was used to support the findings on joint specificity of GDF5 associations with human joint abnormalities, human DDH and knee osteoarthritis, respectively. The analyses in the mice focusing on changes in joint morphology provide further evidence that “knee OA risk at GDF5 is governed by shape alternations” and it is clear that the human data are from patients with radiologically/MRI

diagnosed OA. However, it does not appear that the mice were histologically assessed to determine whether the variant affected cartilage damage as a hallmark of OA. The authors are correct in assuming that there are effects of the variants “on key joint-specific morphological features leading to altered biomechanics and stability” (page 16), which would lead to OA, and the influence of subchondral bone changes on cartilage is well accepted. The bone histomorphometry measurements were performed at P30 in the mice with the DDH-related variant and at 1 year of age in the mice with the knee OA-related variant, so the latter might show osteophytes, as another hallmark of OA. In any case, some note about how structural changes due to either variant in bone, which can adapt, would lead inevitably to OA changes in cartilage, which cannot adapt, would be warranted.”

We thank this Reviewer for noting this deficiency. We had originally established a cohort of mice to study the impacts of the single “T” allele on knee OA. We had to cull our mice because of lab closures due to COVID-19. However, we have now be able to perform histology on a relatively smaller cohort of 1 year old rs6060369 “T” replacement and wildtype mice. We have indeed found that single base-pair replacement mice do develop knee OA via histology. This was evident via the presence of mild cartilage tears to severe cartilage loss in the joints of homozygous replacement mice. Interestingly, consistent with our findings on complete *R4* enhancer null mice, not all replacement mice get knee but only a subset show signs. We have now added these data to the manuscript as new panels in Figure 4 (panels **d** and **e**), and new textual descriptions of the findings in the Results section on the rs6060369 “T” variant. We have also updated the figure legend and included all relevant updates to the Methods section. We are exceptionally pleased by these findings, as they are quite remarkable.

“2. Page 9, Fig. 3a, b, and d: It should be mentioned in the text (although indicated in the legend) that the human chondrocytes used for the ChIP assays were a chondrocyte cell line (T/C-28a2). Upon first reading, since this followed immediately after reference to DDH patients, it appeared that primary chondrocytes from patients might have been used. Although it is well known that primary cells do not generally provide sufficient numbers of cells for ChIP assays, it might be mentioned in the Methods as a justification for using the cell line.”

We have now updated the new text to clearly state that the human ChIP experiments were performed on T/C-28a2 chondrocyte cells *in vitro*. We have also updated the Methods providing a justification for this line.

Overall, we once again thank Reviewer #3 for their terrific insights!

Reviewers' Comments:

Reviewer #3:

Remarks to the Author:

Many thanks to the authors for responding to my additional questions by providing additional data showing histological cartilage damage and OARSI scores in the mice and the clarifications in the text.

Point-by-Point Response to Reviewers (Final Revision)

We thank all Reviewers for their detailed examination of our work and providing excellent constructive comments. All three reviewer had no additional inquiries.

Reviewer #1:

“The authors addressed my comments satisfactorily, the paper is clearer, more focused and improved.”

We thank Reviewer #1 and their critical assessment of our work. Your insights have significantly strengthened the manuscript.

Reviewer #2:

“My queries have been fully addressed in the revised submission materials.”

We thank Reviewer #2 and their critical assessment of our work. Your insights have significantly strengthened the manuscript.

Reviewer #3:

“Many thanks to the authors for responding to my additional questions by providing additional data showing histological cartilage damage and OARSI scores in the mice and the clarifications in the text.”

We thank Reviewer #3 for the kind words and providing excellent comments.